# SECOND-ORDER REGRESSION MODELS EXHIBIT PROGRESSIVE SHARPENING TO THE EDGE OF STABILITY

## ABSTRACT

Recent studies of gradient descent with large step sizes have shown that there is often a regime with an initial increase in the largest eigenvalue of the loss Hessian (progressive sharpening), followed by a stabilization of the eigenvalue near the maximum value which allows convergence (edge of stability). These phenomena are intrinsically non-linear and do not happen for models in the constant Neural Tangent Kernel (NTK) regime, for which the predictive function is approximately linear in the parameters. As such, we consider the next simplest class of predictive models, namely those that are quadratic in the parameters, which we call second-order regression models. For quadratic objectives in two dimensions, we prove that this second-order regression model exhibits progressive sharpening of the NTK eigenvalue towards a value that differs slightly from the edge of stability, which we explicitly compute. In higher dimensions, the model generically shows similar behavior, even without the specific structure of a neural network, suggesting that progressive sharpening and edge-of-stability behavior aren't unique features of neural networks, and could be a more general property of discrete learning algorithms in high-dimensional non-linear models.

## 1 INTRODUCTION

A recent trend in the theoretical understanding of deep learning has focused on the *linearized* regime, where the Neural Tangent Kernel (NTK) controls the learning dynamics (Jacot et al., 2018; Lee et al., 2019). The NTK describes learning dynamics of all networks over short enough time horizons, and can describe the dynamics of wide networks over large time horizons. In the NTK regime, there is a function-space ODE which allows for explicit characterization of the network outputs (Jacot et al., 2018; Lee et al., 2019; Yang, 2021). This approach has been used across the board to gain insights into wide neural networks, but it suffers a major limitation: the model is linear in the parameters, so it describes a regime with relatively trivial dynamics that cannot capture feature learning and cannot accurately represent the types of complex training phenomena often observed in practice.

While other large-width scaling regimes can preserve some non-linearity and allow for certain types of feature learning (Bordelon & Pehlevan, 2022; Yang et al., 2022), such approaches tend to focus on the small learning-rate or continuous-time dynamics. In contrast, recent empirical work has highlighted a number of important phenomena arising from the non-linear discrete dynamics in training practical networks with large learning rates (Neyshabur et al., 2017; Gilmer et al., 2022; Ghorbani et al., 2019; Foret et al., 2022). In particular, many experiments have shown the tendency for networks to display *progressive sharpening* of the curvature towards the *edge of stability*, in which the maximum eigenvalue of the loss Hessian increases over the course of training until it stabilizes at a value equal to roughly two divided by the learning rate, corresponding to the largest eigenvalue for which gradient descent would converge in a quadratic potential (Wu et al., 2018; Giladi et al., 2020; Cohen et al., 2022b;a).

In order to build a better understanding of this behavior, we introduce a class of models which display all the relevant phenomenology, yet are simple enough to admit numerical and analytic understanding. In particular, we propose a simple *quadratic regression model* and corresponding quartic loss function which fulfills both these goals. We prove that under the right conditions, this simple model shows both progressive sharpening *and* edge-of-stability behavior. We then empirically analyze a more general model which shows these behaviors *generically* in the large datapoint, large model

limit. Finally, we conduct a numerical analysis on the properties of a real neural network and use tools from our theoretical analysis to show that edge-of-stability behavior "in the wild" shows some of the same patterns as the theoretical models.

## 2 BASIC QUARTIC LOSS FUNCTION

### 2.1 MODEL DEFINITION

We consider the optimization of the quadratic loss function $\mathcal{L}(\boldsymbol{\theta}) = z^2/2$, where $z$ a quadratic function on the $P \times 1$-dimensional parameter vector $\boldsymbol{\theta}$ and $\mathbf{Q}$ is a $P \times P$ symmetric matrix:

$$z = \frac{1}{2}\left[\boldsymbol{\theta}^\top \mathbf{Q}\boldsymbol{\theta} - E\right] . \tag{1}$$

This can be interpreted either as a model in which the predictive function is quadratic in the input parameters, or as a second-order approximation to a more complicated non-linear function such as a deep network. In this objective, the gradient flow (GF) dynamics with scaling factor $\eta$ is given by

$$\dot{\boldsymbol{\theta}} = -\eta \nabla_{\boldsymbol{\theta}} \mathcal{L} = \eta z \frac{\partial z}{\partial \boldsymbol{\theta}} = \frac{\eta}{2}\left[\boldsymbol{\theta}^\top \mathbf{Q}\boldsymbol{\theta} - E\right]\mathbf{Q}\boldsymbol{\theta} . \tag{2}$$

It is useful to re-write the dynamics in terms of $z$ and the $1 \times P$-dimensional Jacobian $\mathbf{J} = \partial z / \partial \boldsymbol{\theta}$:

$$\dot{z} = -\eta(\mathbf{J}\mathbf{J}^\top)z, \quad \dot{\mathbf{J}} = -2\eta z \mathbf{Q}\mathbf{J} . \tag{3}$$

The curvature is a scalar, described by the neural tangent kernel (NTK) $\mathbf{J}\mathbf{J}^\top$. In these coordinates, we have $E = \mathbf{J}\mathbf{Q}^+\mathbf{J}^\top - 2z$, where $\mathbf{Q}^+$ denotes the Moore-Penrose pseudoinverse.

The GF equations can be simplified by two transformations. First, we transform to $\tilde{z} = \eta z$ and $\tilde{\mathbf{J}} = \eta^{1/2}\mathbf{J}$. Next, we rotate $\boldsymbol{\theta}$ so that $\mathbf{Q}$ is diagonal. This is always possible since $\mathbf{Q}$ is symmetric. Since the NTK is given by $\mathbf{J}\mathbf{J}^\top$, this rotation preserves the dynamics of the curvature. Let $\omega_1 \geq \ldots \geq \omega_P$ be the eigenvalues of $\mathbf{Q}$, and $\mathbf{v}_i$ be the associated eigenvectors (in case of degeneracy, one can pick any basis). We define $\tilde{J}(\omega_i) = \tilde{\mathbf{J}}\mathbf{v}_i$, the projection of $\tilde{\mathbf{J}}$ onto the $i$th eigenvector. Then the gradient flow equations can be written as:

$$\frac{d\tilde{z}}{dt} = -\tilde{z}\sum_{i=1}^{P}\tilde{J}(\omega_i)^2, \quad \frac{d\tilde{J}(\omega_i)^2}{dt} = -2\tilde{z}\omega_i\tilde{J}(\omega_i)^2 . \tag{4}$$

The first equation implies that $\tilde{z}$ does not change sign under GF dynamics. Modes with positive $\omega_i\tilde{z}$ decrease the curvature, and those with negative $\omega_i\tilde{z}$ increase the curvature.

In order to study edge-of-stability behavior, we need initializations which allow the curvature ($\mathbf{J}\mathbf{J}^\top$ in this case) to increase over time - a phenomenon known as *progressive sharpening*. Progressive sharpening has been shown to be ubiquitous in machine learning models (Cohen et al., 2022a), so any useful phenomenological model should show it as well. One such initialization for this quadratic regression model is $\omega_1 = -\omega$, $\omega_2 = \omega$, $\tilde{J}(\omega_1) = \tilde{J}(\omega_2)$. This initialization (and others) show progressive sharpening at all times.

### 2.2 GRADIENT DESCENT

We are interested in understanding the *edge-of-stability* (EOS) behavior in this model: gradient descent (GD) trajectories where the maximum eigenvalue of the NTK, $\mathbf{J}\mathbf{J}^\top$, remains close to the critical value $2/\eta$. We define edge of stability with respect to the maximum NTK eigenvalue instead of the maximum loss Hessian eigenvalue from Cohen et al. (2022a). We will prove this form of EOS in our simpler models, and find that it holds empirically in more complex models. See Appendix A.1 for further discussion.

When $\mathbf{Q}$ has both positive and negative eigenvalues, the loss landscape is the square of a hyperbolic parabaloid (Figure 1, left). As suggested by the gradient flow analysis, this causes some trajectories to increase their curvature before convergence. This causes the final curvature to depend on both the initialization and learning rate. One of the challenges in analyzing the gradient descent (GD) dynamics is that they rapidly and heavily oscillate around minima for large learning rates. One

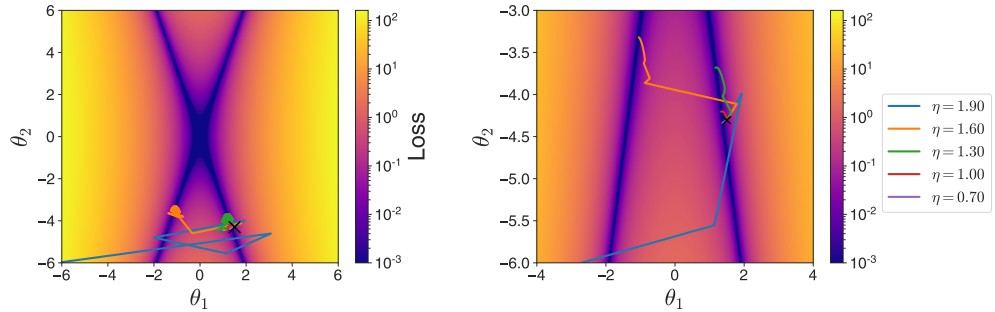

Figure 1: Quartic loss landscape $\mathcal{L}(\cdot)$ as a function of the parameters $\boldsymbol{\theta}$, where $D = 2, E = 0$ and $\mathbf{Q}$ has eigenvalues 1 and $-0.1$. The GD trajectories (initialized at $(1.5, -4.32)$, marked with an x) converge to minima with larger curvature than at initialization and therefore show progressive sharpening (left). The two-step dynamics, in which we consider only even iteration numbers, exhibit fewer oscillations near the edge of stability (right).

way to mitigate this issue is to consider only every other step (Figure 1, right). We will use this observation to analyze the gradient descent (GD) dynamics directly to find configurations where these trajectories show edge-of-stability behavior.

In the eigenbasis coordinates, the gradient descent equations are

$$\tilde{z}_{t+1} - \tilde{z}_t = -\tilde{z}_t \sum_{i=1}^{P} \tilde{J}(\omega_i)_t^2 + \frac{1}{2}(\tilde{z}_t^2) \sum_{i=1}^{P} \omega_i \tilde{J}(\omega_i)_t^2 \tag{5}$$

$$\tilde{J}(\omega_i)_{t+1}^2 - \tilde{J}(\omega_i)_t^2 = -\tilde{z}_t \omega_i (2 - \tilde{z}_t \omega_i) \tilde{J}(\omega_i)_t^2 \text{ for all } 1 \leq i \leq P. \tag{6}$$

We'll find it convenient in the following to write the dynamics in terms of weighted averages of $\tilde{J}(\omega_i)^2$ instead of the modes $\tilde{J}(\omega_i)$:

$$T(\alpha) = \sum_{i=1}^{P} \omega_i^\alpha \tilde{J}(\omega_i)^2. \tag{7}$$

The dynamical equations become:

$$\tilde{z}_{t+1} - \tilde{z}_t = -\tilde{z}_t T_t(0) + \frac{1}{2}(\tilde{z}_t^2) T_t(1) \tag{8}$$

$$T_{t+1}(k) - T_t(k) = -\tilde{z}_t (2T_t(k+1) - \tilde{z}_t T_t(k+2)). \tag{9}$$

If $\mathbf{Q}$ is invertible, then we have $E = T_t(-1) - 2\tilde{z}_t$. Note that by definition $T_t(0) = \eta \mathbf{J}_t \mathbf{J}_t^\top$ is the (rescaled) NTK. edge-of-stability behavior corresponds to dynamics which keep $T_t(0)$ near the value 2 as $\tilde{z}_t$ goes to 0.

### 2.2.1 REDUCTION TO CATAPULT DYNAMICS

If the eigenvalues of $\mathbf{Q}$ are $\{-\omega, \omega\}$, and $E = 0$, the model becomes equivalent to a single hidden layer linear network with one training datapoint (Appendix A.2) - also known as the catapult phase dynamics. This model doesn't exhibit sharpening or edge-of-stability behavior (Lewkowycz et al., 2020). We will analyze this model in our $\tilde{z} - T(0)$ variables as a warmup, with an eye towards analyzing a different parameter setting which does show sharpening and edge of stability.

We assume without loss of generality that the eigenvalues are $\{-1, 1\}$ - which can be accomplished by rescaling $\tilde{z}$. The loss function is then the square of a hyperbolic paraboloid. Since there are only 2 variables, we can rewrite the dynamics in terms of $\tilde{z}$ and the curvature $T(0)$ only (Appendix B.1):

$$\tilde{z}_{t+1} - \tilde{z}_t = -\tilde{z}_t T_t(0) + \frac{1}{2}(\tilde{z}_t^2)(2\tilde{z}_t + E) \tag{10}$$

$$T_{t+1}(0) - T_t(0) = -2\tilde{z}_t(2\tilde{z}_t + E) + z_t^2 T_t(0). \tag{11}$$

For $E = 0$, we can see that $\text{sign}(\Delta T(0)) = \text{sign}(T_t(0) - 4)$, as in Lewkowycz et al. (2020) - so convergence requires strictly decreasing curvature. For $E \neq 0$, there is a region where the curvature can increase (Appendix B.1). However, there is still no edge-of-stability behavior - there is no set of initializations which starts with $\lambda_{\max}$ far from $2/\eta$, which ends up near $2/\eta$. In contrast, we will show that asymmetric eigenvalues can lead to EOS behavior.

### 2.2.2 EDGE OF STABILITY REGIME

In this section, we consider the case in which $\mathbf{Q}$ has two eigenvalues - one of which is large and positive, and the other one small and negative. Without loss of generality, we assume that the largest eigenvalue of $\mathbf{Q}$ is 1. We denote the second eigenvalue by $-\epsilon$, for $0 < \epsilon \leq 1$. With this notation we can write the dynamical equations (Appendix B.1) as

$$\tilde{z}_{t+1} - \tilde{z}_t = -\tilde{z}_t T_t(0) + \frac{1}{2}(\tilde{z}_t^2)((1 - \epsilon)T_t(0) + \epsilon(2\tilde{z}_t + E)) \tag{12}$$

$$T_{t+1}(0) - T_t(0) = -2\tilde{z}_t(\epsilon(2\tilde{z}_t + E) + (1 - \epsilon)T_t(0)) + \tilde{z}_t^2\left[T_t(0) + \epsilon(\epsilon - 1)(T_t(0) - E - 2\tilde{z}_t)\right]. \tag{13}$$

For small $\epsilon$, there are trajectories where $\lambda_{\max}$ is initially away from $2/\eta$ but converges towards it (Figure 2, left) - in other words, EOS behavior. We used a variety of step sizes $\eta$ but initialized at pairs initialized at pairs $(\eta z_0, \eta T_0(0))$ to show the universality of the $\tilde{z}$-$T(0)$ coordinates.

In order to quantitatively understand the progressive sharpening and edge of stability, it is useful to look at the two-step dynamics. One additional motivation for studying the two-step dynamics follows from the analysis of gradient descent on linear least squares (i.e., linear model) with a large step size $\lambda$. For every coordinate $\tilde{\theta}$, the one-step and two-step dynamics are

$$\tilde{\theta}_{t+1} - \tilde{\theta}_t = -\lambda\tilde{\theta}_t \quad \text{and} \quad \tilde{\theta}_{t+2} - \tilde{\theta}_t = (1 - \lambda)^2\tilde{\theta}_t \qquad \text{(GD in quadratic potential)}. \tag{14}$$

While the dynamics converge for $\lambda < 2$, if $\lambda > 1$ the one-step dynamics oscillate when approaching minimum, whereas the the two-step dynamics maintain the sign of $\tilde{\theta}$ and the trajectories exhibit no oscillations.

Likewise, plotting every other iterate in the two parameter model more clearly demonstrates the phenomenology. For small $\epsilon$, the dynamics shows the distinct phases described in (Li et al., 2022): an initial increase in $T(0)$, a slow increase in $\tilde{z}$, then a decrease in $T(0)$, and finally a slow decrease of $\tilde{z}$ while $T(0)$ remains near 2 (Figure 2, middle).

Unfortunately, the two-step version of the dynamics defined by Equations 12 and 13 are more complicated – they are 3rd order in $T(0)$ and 9th order in $\tilde{z}$; see Appendix B.2 for a more detailed discussion. However we can still analyze the dynamics as $\tilde{z}$ goes to 0. In order to understand the mechanisms of the EOS behavior, it is useful to understand the *nullclines* of the two step dynamics. The nullcline $f_{\tilde{z}}(\tilde{z})$ of $\tilde{z}$ and $f_T(\tilde{z})$ of $T(0)$ are defined implicitly by

$$(\tilde{z}_{t+2} - \tilde{z}_t)(\tilde{z}, f_{\tilde{z}}(\tilde{z})) = 0, \quad (T_{t+2}(0) - T_t(0))(\tilde{z}, f_T(\tilde{z})) = 0 \tag{15}$$

where $\tilde{z}_{t+2} - \tilde{z}_t$ and $T_{t+2}(0) - T_t(0)$ are the aforementioned high order polynomials in $\tilde{z}$ and $T(0)$. Since these polynomials are cubic in $T(0)$, there are three possible solutions as $\tilde{z}$ goes to 0. We are particularly interested in the solution that goes through $\tilde{z} = 0$, $T(0) = 2$ - that is, the critical point corresponding to EOS.

Calculations detailed in Appendix B.2 show that the distance between the two nullclines is linear in $\epsilon$, so they become close as $\epsilon$ goes to 0. (Figure 2, middle). In addition, the trajectories stay near $f_{\tilde{z}}$ - which gives rise to EOS behavior. This suggests that the dynamics are slow near the nullclines, and trajectories appear to be approaching an attractor. We can find the structure of the attractor by changing variables to $y_t \equiv T_t(0) - f_{\tilde{z}}(\tilde{z}_t)$ - the distance from the $\tilde{z}$ nullcline. To lowest order in $\tilde{z}$ and $y$, the two-step dynamical equations become (Appendix B.3):

$$\tilde{z}_{t+2} - \tilde{z}_t = 2y_t\tilde{z}_t + O(y_t^2\tilde{z}_t) + O(y_t\tilde{z}_t^2) \tag{16}$$

$$y_{t+2} - y_t = -2(4 - 3\epsilon + 4\epsilon^2)y_t\tilde{z}_t^2 - 4\epsilon\tilde{z}_t^2 + \epsilon O(\tilde{z}_t^3) + O(y^2\tilde{z}_t^2) \tag{17}$$

We immediately see that $\tilde{z}$ changes slowly for small $y$ - since we chose coordinates where $\tilde{z}_{t+2} - \tilde{z}_t = 0$ when $y = 0$. We can also see that $y_{t+2} - y_t$ is $O(\epsilon)$ for $y_t = 0$ - so for small $\epsilon$, the $y$ dynamics

is slow too. Moreover, we see that the coefficient of the $\epsilon \tilde{z}_t^2$ term is negative - the changes in $\tilde{z}$ tend to drive $y$ (and therefore $T(0)$) to decrease. The coefficient of the $y_t$ term is negative as well; the dynamics of $y$ tends to be contractive. The key is that the contractive behavior takes $y$ to an $O(\epsilon)$ fixed point at a rate proportional to $\tilde{z}^2$, while the dynamics of $\tilde{z}$ are proportional to $\epsilon$. This suggests a separation of timescales if $\tilde{z}^2 \gg \epsilon$, where $y$ first equilibrates to a fixed value, and then $\tilde{z}$ converges to 0 (Figure 2, right). This intuition for the lowest order terms can be formalized, and gives us a prediction of $\lim_{t\to\infty} y_t = -\epsilon/2$, confirmed numerically in the full model (Appendix B.5).

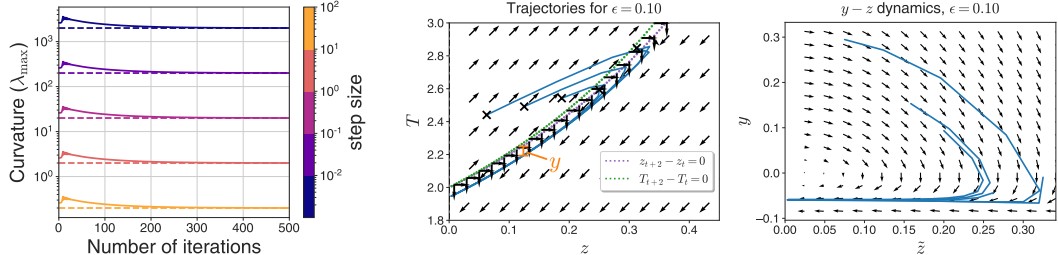

Figure 2: For small $\epsilon$, two-eigenvalue model shows EOS behavior for various step sizes ($\epsilon = 5 \cdot 10^{-3}$, left). Trajectories are the same up to scaling because corresponding rescaled coordinates $\tilde{z}$ and $T(0)$ are the same at initialization. Plotting every other iterate, we see that for a variety of initializations (black x's), trajectories in $\tilde{z} - T(0)$ space stay near the nullcline $(\tilde{z}, f_{\tilde{z}}(\tilde{z}))$ - the curve where $\tilde{z}_{t+2} - \tilde{z}_t = 0$ (middle). Changing variables to $y = T(0) - f_{\tilde{z}}(\tilde{z})$ shows quick concentration to a curve of near-constant, small, negative $y$ (right).

We can prove the following theorem about the long-time dynamics of $\tilde{z}$ and $y$ when the higher order terms are included (Appendix B.4):

**Theorem 2.1.** *There exists an $\epsilon_c > 0$ such that for a quadratic regression model with $E = 0$ and eigenvalues $\{-\epsilon, 1\}$, $\epsilon \le \epsilon_c$, there exists a neighborhood $U \subset \mathbb{R}^2$ and interval $[\eta_1, \eta_2]$ such that for initial $\boldsymbol{\theta} \in U$ and learning rate $\eta \in [\eta_1, \eta_2]$, the model displays edge-of-stability behavior:*

$$2/\eta - \delta_\lambda \le \lim_{t\to\infty} \lambda_{\max} \le 2/\eta, \tag{18}$$

*for $\delta_\lambda$ of $O(\epsilon)$. This neighborhood corresponds to the inverse image of the $\tilde{z} - y$ space region $[0, \tilde{z}_c) \times [0, y_c)$, for $\epsilon$-independent $\tilde{z}_c$ and $y_c$.*

Therefore, unlike the catapult phase model, the small $\epsilon$ provably has EOS behavior - whose mechanism is well-understood by the $\tilde{z} - y$ coordinate transformation.

## 3 QUADRATIC REGRESSION MODEL

### 3.1 GENERAL MODEL

While the model defined in Equation 1 provable displays edge-of-stability behavior, it required tuning of the eigenvalues of $\mathbf{Q}$ to demonstrate it. We can define a more general model which exhibits edge-of-stability behavior with less tuning. We define the *quadratic regression model* as follows. Given a $P$-dimensional parameter vector $\boldsymbol{\theta}$, the $D$-dimensional output vector $\mathbf{z}$ is given by

$$\mathbf{z} = \mathbf{y} + \mathbf{G}^\top \boldsymbol{\theta} + \frac{1}{2}\mathbf{Q}(\boldsymbol{\theta}, \boldsymbol{\theta}). \tag{19}$$

Here $\mathbf{y}$ is a $D$-dimensional vector, $\mathbf{G}$ is a $D \times P$-dimensional matrix, and $\mathbf{Q}$ is a $D \times P \times P$-dimensional tensor symmetric in the last two indices - that is, $\mathbf{Q}(\cdot, \cdot)$ takes two $P$-dimensional vectors as input, and outputs a $D$-dimensional vector verifying $\mathbf{Q}(\boldsymbol{\theta}, \boldsymbol{\theta})_\alpha = \boldsymbol{\theta}^\top \mathbf{Q}_\alpha \boldsymbol{\theta}$. If $\mathbf{Q} = \mathbf{0}$, the model corresponds to linearized learning (as in the NTK regime). When $\mathbf{Q} \ne \mathbf{0}$, we obtain the first correction to NTK regime. We note that:

$$\mathbf{G}_{\alpha i} = \frac{\partial \mathbf{z}_\alpha}{\partial \boldsymbol{\theta}_i}\bigg|_{\boldsymbol{\theta}=0}, \quad \mathbf{Q}_{\alpha i j} = \frac{\partial^2 \mathbf{z}_\alpha}{\partial \boldsymbol{\theta}_i \partial \boldsymbol{\theta}_j}, \to \mathbf{J} = \mathbf{G} + \mathbf{Q}(\boldsymbol{\theta}, \cdot), \tag{20}$$

for the $D \times P$ dimensional Jacobian $\mathbf{J}$. For $D = 1$, we recover the model of Equation 1. In the remainder of this section, we will study the limit as $D$ and $P$ increase with fixed ratio $D/P$.

The quadratic regression model corresponds to a model with a constant second derivative with respect to parameter changes - or a second order expansion of a more complicated ML model. Quadratic expansions of shallow MLPs have been previously studied (Bai & Lee, 2020; Zhu et al., 2022), and the perturbation theory for small $\mathbf{Q}$ is studied in Roberts et al. (2022). Other related models are detailed in Appendix A. We will provide evidence that even random, unstructured quadratic regression models lead to EOS behavior.

## 3.2 GRADIENT FLOW DYNAMICS

We will focus on training with squared loss $\mathcal{L}(\mathbf{z}) = \frac{1}{2} \sum_\alpha \mathbf{z}_\alpha^2$. We begin by considering the dynamics under gradient flow (GF):

$$\dot{\boldsymbol{\theta}} = -\frac{\partial \mathcal{L}(\mathbf{z})}{\partial \boldsymbol{\theta}} = -\mathbf{J}^\top \mathbf{z} \,. \tag{21}$$

We can write the dynamics in the output space $\mathbf{z}$ and the Jacobian $\mathbf{J}$ as

$$\dot{\mathbf{z}} = \mathbf{J}\dot{\boldsymbol{\theta}} = -\mathbf{J}\mathbf{J}^\top \mathbf{z}, \quad \dot{\mathbf{J}} = -\mathbf{Q}(\mathbf{J}^\top \mathbf{z}, \cdot) \tag{22}$$

When $\mathbf{Q} = \mathbf{0}$ (linearized/NTK regime), $\mathbf{J}$ is constant, the dynamics are then linear in $\mathbf{z}$, and are controlled by the eigenstructure of $\mathbf{J}\mathbf{J}^\top$, the empirical NTK. In this regime there is no EOS behavior.

We are interested in settings where progressive sharpening occurs under GF. We can study the dynamics of the maximum eigenvalue $\lambda_{\max}$ of $\mathbf{J}\mathbf{J}^\top$ at early times for random initializations. In Appendix C.1, we prove the following theorem:

**Theorem 3.1.** *Let $\mathbf{z}$, $\mathbf{J}$, and $\mathbf{Q}$ be initialized with i.i.d. elements with zero mean and variances $\sigma_z^2$, $\sigma_J^2$, and 1 respectively, with distributions invariant to rotation in data and parameter space, and have finite fourth moments. Let $\lambda_{\max}$ be the largest eigenvalue of $\mathbf{J}\mathbf{J}^\top$. In the limit of large $D$ and $P$, with fixed ratio $D/P$, at initialization we have*

$$\mathrm{E}[\dot{\lambda}_{\max}(0)] = 0, \ \mathrm{E}[\ddot{\lambda}_{\max}(0)]/\mathrm{E}[\lambda_{\max}(0)] = \sigma_z^2 \tag{23}$$

*where $\mathrm{E}$ denotes the expectation over $\mathbf{z}$, $\mathbf{J}$, and $\mathbf{Q}$ at initialization.*

Much like in the $D = 1$ case, Theorem 3.1 suggests that it is easy to find initializations that show progressive sharpening - and increasing $\sigma_z$ makes sharpening more prominent.

## 3.3 GRADIENT DESCENT DYNAMICS

We now consider finite-step size gradient descent (GD) dynamics. The dynamics for $\boldsymbol{\theta}$ are given by:

$$\boldsymbol{\theta}_{t+1} = \boldsymbol{\theta}_t - \eta \mathbf{J}_t^\top \mathbf{z}_t \,. \tag{24}$$

In this setting, the dynamic equations can be written as

$$\mathbf{z}_{t+1} - \mathbf{z}_t = -\eta \mathbf{J}_t \mathbf{J}_t^\top \mathbf{z}_t + \frac{1}{2}\eta^2 \mathbf{Q}(\mathbf{J}_t^\top \mathbf{z}_t, \mathbf{J}_t^\top \mathbf{z}_t) \tag{25}$$

$$\mathbf{J}_{t+1} - \mathbf{J}_t = -\eta \mathbf{Q}(\mathbf{J}_t^\top \mathbf{z}_t, \cdot) \,. \tag{26}$$

If $\mathbf{Q} = \mathbf{0}$, the dynamics reduce to discrete gradient descent in a quadratic potential - which converges iff $\lambda_{\max} < 2/\eta$.

One immediate question is: when does the $\eta^2$ in Equation 25 affect the dynamics? Given that it scales with higher powers of $\eta$ and $\mathbf{z}$ than the first term, we can conjecture that the ratio of the magnitudes of the terms, $r_{NL}$, is proportional to $||\mathbf{z}||_2$ and $\eta$. A calculation in Appendix C.2 shows that, for random rotationally invariant initializations, we have:

$$r_{NL} \equiv \left( \frac{\mathrm{E}[||\frac{1}{2}\eta^2 \mathbf{Q}(\mathbf{J}_0^\top \mathbf{z}_0, \mathbf{J}_0^\top \mathbf{z}_0)||_2^2]}{\mathrm{E}[||\eta \mathbf{J}_0 \mathbf{J}_0^\top \mathbf{z}_0||_2^2]} \right)^{1/2} = \frac{1}{2}\eta \sigma_z D \,, \tag{27}$$

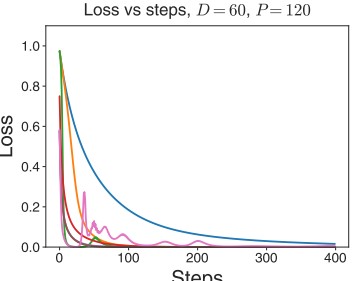
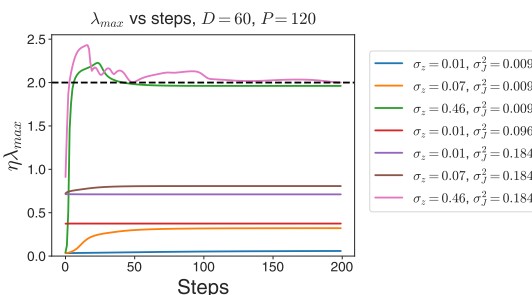

Figure 3: Gradient descent dynamics in the quadratic regression model. As $\mathbf{z}$ initialization variance $\sigma_z^2$ increases, so does the curvature $\lambda_{\max}$ upon convergence. As sharpening drives $\eta\lambda_{\max}$ near 2, larger $\sigma_z$ allows for non-linear effects to induce edge-of-stability behavior (right). Resulting loss trajectories are non-monotonic but still converge to 0 (left).

where as before the expectation is taken over the initialization of $\mathbf{z}$, $\mathbf{J}$, and $\mathbf{Q}$. This suggests that increasing the learning rate increases the deviation of the dynamics from GF (which is obvious), but increasing $\|\mathbf{z}\|$ *also* increases the deviation from GF.

We can see this phenomenology in the dynamics of the GD equations (Figure 3). Here we plot different trajectories for random initializations of the type in Theorem 3.1 with $D = 60$, $P = 120$, and $\eta = 1$. As $\sigma_z$ increases, so does the curvature $\lambda_{\max}$ (as suggested by Theorem 3.1), and when $\sigma_z$ is $O(1)$, the dynamics is non-linear (as predicted by $r_{NL}$) and EOS behavior emerges. This suggests that the second term in Equation 25 is crucial for the stabilization of $\lambda_{max}$.

We can confirm this more generally by initializing over various $\eta$, $D$, $P$, $\sigma_z$, and $\sigma_J$ over multiple seeds, and plotting the resulting phase diagram of the final $\lambda_{\max}$ reached. We can simplify the plotting with some rescaling of parameters and initializations. For example, in the rescaled variables

$$\tilde{\mathbf{z}} = \eta\mathbf{z}, \ \tilde{\mathbf{J}} = \eta^{1/2}\mathbf{J} \,, \tag{28}$$

the dynamics are equivalent to Equations 25 and 26 with $\eta = 1$. As in the $\tilde{z} - T(0)$ model of Equations 8–9, $\lambda_{\max}$ in the rescaled coordinates is equivalent to $\eta\lambda_{\max}$ in the unscaled coordinates. We can also define rescaled initializations for $\mathbf{z}$ and $\mathbf{J}$. If we set

$$\sigma_z = \tilde{\sigma}_z/D, \ \sigma_J = \tilde{\sigma}_J/\left(DP\right)^{1/4} \,, \tag{29}$$

then we have $r_{NL} = \tilde{\sigma}_z$ which allows for easier comparison across $(D, P)$ pairs.

Using this initialization scheme, we can plot the final value of $\lambda_{\max}$ reached as a function of $\tilde{\sigma}_z$ and $\tilde{\sigma}_J$ for 100 independent random initializations for each $\tilde{\sigma}_z$, $\tilde{\sigma}_J$ pair (Figure 4). We see that the key is for $r_{NL} = \tilde{\sigma}_z$ to be $O(1)$ - corresponding to both progressive sharpening and non-linear dynamics near initialization. In particular, initializations with small $\tilde{\sigma}_J$ values which converge at the EOS correspond to trajectories which first sharpen, and then settle near $\lambda_{\max} = 2/\eta$. Large $\tilde{\sigma}_z$ and large $\tilde{\sigma}_J$ dynamics diverge. There is a small band of initial $\tilde{\sigma}_J$ over a wide range of $\tilde{\sigma}_z$ which have final $\lambda_{\max} \approx 2/\eta$; these correspond to models initialized near the EOS, which stay near it.

This suggests that progressive sharpening and edge of stability aren't uniquely features of neural network models, and could be a more general property of learning in high-dimensional, non-linear models.

## 4 CONNECTION TO REAL WORLD MODELS

In this section we examine how representative is the proposed model and the developed theory to the behavior of "real world" models. Following Cohen et al. (2022a), we trained a 2-hidden layer $\tanh$ network using the squared loss on 5000 examples from CIFAR10 with learning rate $10^{-2}$ - a setting which shows edge of stability behavior. Close to the onset of EOS, we approximately computed $\lambda_1$, the largest eigenvalue of $\mathbf{JJ}^\top$, and its corresponding eigenvector $\mathbf{v}_1$ using a Lanczos method (Ghorbani et al., 2019; Novak et al., 2019). We use $\mathbf{v}_1$ to compute $z_1 = \mathbf{v}_1^\top \mathbf{z}$, where $\mathbf{z}$ is the vector of residuals $f(\mathbf{X}, \boldsymbol{\theta}) - \mathbf{Y}$ for neural network function $f$, training inputs $\mathbf{X}$, labels $\mathbf{Y}$, and

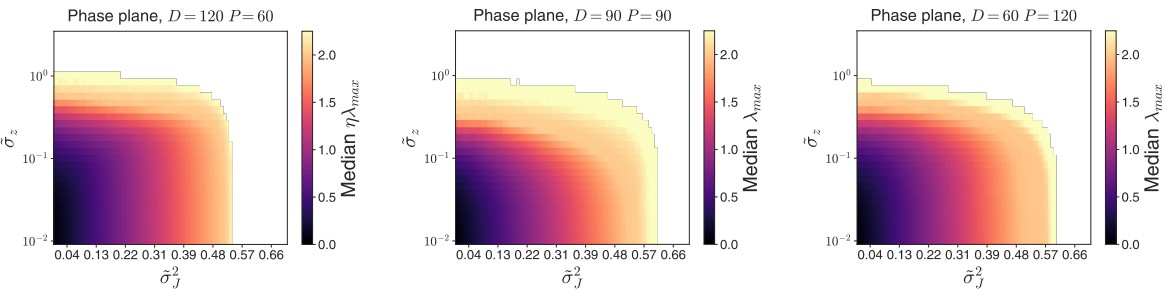

Figure 4: $\tilde{\sigma}_z/\tilde{\sigma}_J^2$ phase planes for quadratic regression models, for various $D$ and $P$. Models were initialized with 100 random seeds for each $\tilde{\sigma}_z, \tilde{\sigma}_J$ pair and iterated until convergence. For each pair $\tilde{\sigma}_z, \tilde{\sigma}_J^2$ we plot the median $\lambda_{\max}$ of the NTK $\mathbf{J}^\top \mathbf{J}$. For intermediate $\tilde{\sigma}_z$, where both sharpening and non-linear $\mathbf{z}$ dynamics occur, trajectories tend to converge so $\lambda_{\max}$ of the NTK is near $2/\eta$ (EOS).

parameters $\boldsymbol{\theta}$. The EOS behavior in the NTK is similar to the EOS behavior defined with respect to the full Hessian in Cohen et al. (2022a) (Figure 5, left and right). Once again, plotting the trajectories at every other step gets rid of the high frequency oscillations (Figure 5, middle). Unlike the $D = 1$, $P = 2$ model, there are multiple crossings of the critical line $\lambda_{\max} = 2/\eta$ line.

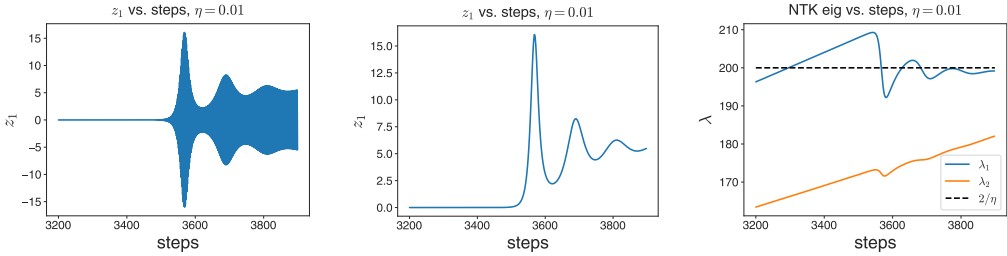

Figure 5: A FCN trained on CIFAR shows multiple cycles of sharpening and edge-of-stability behavior. $z_1$, the projection of the training set residuals $f(\mathbf{X}, \boldsymbol{\theta}) - \mathbf{Y}$ onto the top NTK eigenmode $\mathbf{v}_1$, increases in magnitude and oscillates around 0 (left). Plotting dynamics every two steps removes high frequency oscillations (middle). The largest eigenvalue $\lambda_1$ crosses the edge of stability multiple times, but the second largest eigenvalue $\lambda_2$ remains below the edge of stability.

There is evidence that low-dimensional features of a quadratic regression model could be used to explain some aspects of EOS behavior. We empirically compute the the second derivative of the output $f(\mathbf{x}, \boldsymbol{\theta})$ by automatic differentiation. We denote by $\mathbf{Q}(\cdot, \cdot)$ the resulting tensor. We can use matrix-vector products to compute the spectrum of the matrix $\mathbf{Q}_1 \equiv \mathbf{v}_1 \cdot \mathbf{Q}(\cdot, \cdot)$, which is projection of the output of $\mathbf{Q}$ in the $\mathbf{v}_1$ direction, without instantiating $\mathbf{Q}$ in memory (Figure 6, left). This figure reveals that the spectrum does not shift much from step 3200 to 3900 (the range of our plots). This suggests that $\mathbf{Q}$ doesn't change much as these EOS dynamics are displayed. We can also see that $\mathbf{Q}$ is much larger in the $\mathbf{v}_1$ direction than a random direction.

Let $y$ be defined as $y = \lambda_1 \eta - 2$. Plotting the two-step dynamics of $z_1$ versus $2yz$ we see a remarkable agreement (Figure 6, middle). This is the same form that the dynamics of $\tilde{z}$ takes in our simplified model. It can also be found by iterating Equation 25 twice with fixed Jacobian for $y = \lambda_1 \eta - 2$ and discarding terms higher order in $\eta$. This suggests that during this particular EOS behavior, much like in our simplified model the dynamics of the eigenvalue is more important than any rotation in the eigenbasis.

The dynamics of $y$ is more complicated; $y_{t+2} - y_t$ is anticorrelated with $z_1^2$ but there is no low-order functional form in terms of $y$ and $z_1$ (Appendix D.1). We can get some insight into the stabilization by plotting the ratio of $\eta^2 \mathbf{Q}_1(\mathbf{J} z_1 \mathbf{v}_1, \mathbf{J} z_1 \mathbf{v}_1)$ (the non-linear contribution to the $z_1$ dynamics from the $\mathbf{v}_1$ direction) and $\lambda_1 z_1$ (the linearized contribution), and compare it to the dynamics of $y$ (Figure 6, right). The ratio is small during the initial sharpening, but becomes $O(1)$ shortly before the

curvature decreases for the first time. It remains $O(1)$ through the rest of the dynamics. This suggests that the non-linear feedback from the dynamics of the top eigenmode onto itself is crucial to understanding the EOS dynamics.

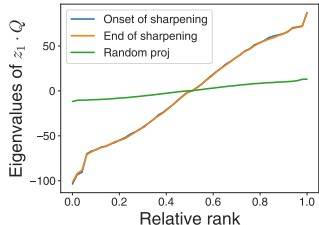 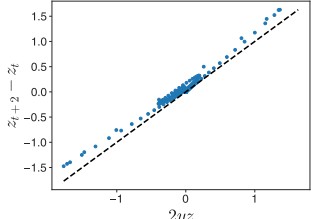 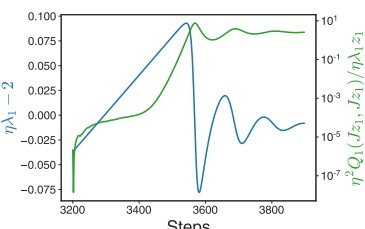

Figure 6: $\mathbf{Q}$ is approximately constant during edge-of-stability dynamics for FCN trained on CIFAR10 (left). Projection onto largest eigendirection $\mathbf{v}_1$ (blue and orange) is larger than projection onto random direction (green). Two step difference $(z_1)_{t+2} - (z_1)_t$ is well approximated by $2z_1 y$ (middle), leading order term of models with fixed eigenbasis. Non-linear dynamical contribution $\eta^2 \mathbf{Q}_1(\mathbf{J}z_1\mathbf{v}_1, \mathbf{J}z_1\mathbf{v}_1)$ is small during sharpening, but becomes large immediately preceding decrease in top eigenvalue (right) - as is the case in the simple model.

## 5 DISCUSSION

### 5.1 LESSONS LEARNED FROM QUADRATIC REGRESSION MODELS

The main lesson to be learned from the quadratic regression models is that behavior like progressive sharpening (for both GF and GD) and edge-of-stability behavior (for GD) may be common features of high-dimensional gradient-based training of non-linear models. Indeed, these phenomena can be revealed in simple settings without any connection to deep learning models: with mild tuning our simplified model, which corresponds to 1 datapoint and 2 parameters can provably show EOS behavior. This combined with the analysis of the CIFAR model suggest that the general mechanism may have a low-dimensional description.

Quadratic approximations of real models quantitatively can capture the early features of EOS behavior (the initial return to $\lambda_{max} < 2/\eta$), but do not necessarily capture the magnitude and period of subsequent oscillations – these require higher order terms (Appendix D.2). Nevertheless, the quadratic approximation does correctly describe much of the qualitative behavior, including the convergence of $\lambda_{max}$ to a limiting two-cycle that oscillates around $2/\eta$, with an average value *below* $2/\eta$. In the simplified two-parameter model, it is possible to analytically predict the final value at convergence, and indeed we find that it deviates slightly from the value $2/\eta$.

A key feature of all the models studied in this work is that looking at every-other iterate (the two-step dynamics) greatly aids in understanding the models theoretically and empirically. Near the edge of stability, this makes the changes in the top eigenmode small. In the simplified model, the slow $\tilde{z}$ dynamics (and related slow $T(0)$ dynamics) allowed for the detailed theoretical analysis; in the CIFAR model, the two-step dynamics is slowly varying in both $z_1$ and $\lambda_{max}$. The quantitative comparisons of these small changes may help uncover any universal mechanisms/canonical forms that explain EOS behavior in other systems and scenarios.

### 5.2 FUTURE WORK

One avenue for future work is to quantitatively understand progressive sharpening and EOS behavior in the quadratic regression model for large $D$ and $P$. In particular, it may be possible to predict the final deviation $2 - \eta \lambda_{max}$ in the edge-of-stability regime as a function of $\sigma_z$, $\sigma_J$, and $D/P$. It would also be useful to understand how higher order terms affect the training dynamics. One possibility is that a small number of statistics of the higher order derivatives of the loss function are sufficient to obtain a better quantitative understanding of the oscillations around $y = 2$.

Finally, our analysis has not touched on the feature learning aspects of the model. In the quadratic regression model, feature learning is encoded in the relationship between $\mathbf{J}$ and $\mathbf{z}$, and in particu-

lar the relationship between $\mathbf{z}$ and the eigenstructure of $\mathbf{JJ}^\top$. Understanding how $\mathbf{Q}$ mediates the dynamics of these two quantities may provide a quantitative basis for understanding feature learning which is complementary to existing theoretical approaches (Roberts et al., 2022; Bordelon & Pehlevan, 2022; Yang et al., 2022).

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

## A CONNECTION TO OTHER MODELS

### A.1 HESSIAN VERSUS NTK MAXIMUM EIGENVALUE

In this work we focus on EOS dynamics of the largest eigenvalue of the NTK, rather than the Hessian as in Cohen et al. (2022a). We note that a version of Theorem 2.1 is true for the maximum Hessian eigenvalue as well. In general, the Hessian can be written as

$$\frac{\partial^2 \mathcal{L}}{\partial \boldsymbol{\theta} \partial \boldsymbol{\theta}'} = \nabla \mathcal{L} \cdot \frac{\partial^2 \mathbf{z}}{\partial \boldsymbol{\theta} \partial \boldsymbol{\theta}'} + \mathbf{J}^{\mathrm{T}} \frac{\partial^2 \mathcal{L}}{\partial \mathbf{z} \partial \mathbf{z}'} \mathbf{J} \tag{30}$$

For squared loss in particular, we have

$$\frac{\partial^2 \mathcal{L}}{\partial \boldsymbol{\theta} \partial \boldsymbol{\theta}'} = \nabla \mathcal{L} \cdot \frac{\partial^2 \mathbf{z}}{\partial \boldsymbol{\theta} \partial \boldsymbol{\theta}'} + \mathbf{J}^{\mathrm{T}} \mathbf{J} \tag{31}$$

As the loss gradient goes to 0, the Hessian eigenvalues approach the eigenvalues of $\mathbf{J}\mathbf{J}^{\mathrm{T}}$ - whose non-zero eigenvalues are the same as those of the empirical NTK $\mathbf{J}\mathbf{J}^{\mathrm{T}}$. Since the theorem involves behavior as $\tilde{z}$ goes to convergence, the maximum NTK and maximum Hessian eigenvalues are equal in the limit, and the same EOS behavior applied in both cases.

For the higher dimensional models (quadratic regression model and fully connected network on CIFAR10), our experiments show that the maximum NTK eigenvalue shows edge of stability behavior. The CIFAR model is the same as the one in Cohen et al. (2022a) which was used to illustrate the edge of stability in terms of the maximum Hessian eigenvalues. Therefore we focused on the NTK version of EOS in our paper, as we found it more amenable to theoretical analysis and explanation.

There are almost certainly cases where EOS behavior is displayed in the Hessian eigenvalues but not the NTK eigenvalues, particularly in cases where the loss is highly non-isotropic in the outputs (that is, $\frac{\partial^2 \mathcal{L}}{\partial \mathbf{z} \partial \mathbf{z}'}$ is far from a multiple of the identity matrix). As pointed out in previous works in these cases even the Hessian-based EOS is more difficult to analyze Cohen et al. (2022a). We leave understanding of EOS with more complicated loss functions for future work.

### A.2 ONE-HIDDEN LAYER LINEAR NETWORK

Consider a one hidden layer network with a scalar output:

$$f(\mathbf{x}) = \mathbf{v}^\top \mathbf{U} \mathbf{x} \tag{32}$$

where $\mathbf{x}$ is an input vector of length $N$, $\mathbf{U}$ is a $K \times N$ dimensional matrix, and $\mathbf{v}$ is a $K$ dimensional vector. We note that

$$\frac{\partial^2 f(\mathbf{x})}{\partial \mathbf{v}_i \partial \mathbf{v}_j} = \frac{\partial^2 f(\mathbf{x})}{\partial \mathbf{U}_{ij} \partial \mathbf{U}_{kl}} = 0, \ \frac{\partial^2 f(\mathbf{x})}{\partial \mathbf{v}_i \partial \mathbf{U}_{jk}} = \delta_{ij} \mathbf{x}_k \tag{33}$$

where $\delta_{ij}$ is the Kroenecker delta. For a fixed training set, this second derivative is constant; therefore, the one-hidden layer linear network is a quadratic regression model of the type studied in Section 3.

In the particular case of a single datapoint $\mathbf{x}$, we can compute the eigenvectors of the $\mathbf{Q}$ matrix. Let $(\mathbf{w}, \mathbf{W})$ be an eigenvector of $\mathbf{Q}$, representing the $\mathbf{v}$ and $\mathbf{U}$ components respectively. The eigenvector equations are

$$\omega\mathbf{w}_i = \mathbf{x}_m\delta_{ij}\mathbf{W}_{jm} \tag{34}$$

$$\omega\mathbf{W}_{jm} = \mathbf{x}_m\delta_{ij}\mathbf{w}_i \tag{35}$$

Simplifying, we have:

$$\omega\mathbf{w} = \mathbf{W}\mathbf{x} \tag{36}$$

$$\omega\mathbf{W} = \mathbf{w}\mathbf{x}^\top \tag{37}$$

We have two scenarios. The first is that $\omega = 0$. In this case, we have $\mathbf{w} = 0$, and $\mathbf{W}$ is a matrix with $\mathbf{x}$ in its nullspace. The latter condition gives us $M$ constraints on $M \times N$ equations - for a total of $M(N-1)$ of our $M(N+1)$ total eigenmodes.

If $\omega \neq 0$, then combining the equations we have the conditions:

$$\omega^2\mathbf{w} = (\mathbf{x}\cdot\mathbf{x})\mathbf{w} \tag{38}$$

$$\omega^2\mathbf{W} = \mathbf{W}\mathbf{x}\mathbf{x}^\top \tag{39}$$

This gives us $\omega = \pm\sqrt{\mathbf{x}\cdot\mathbf{x}}$. We know from Equation 37 that $\mathbf{W}$ is low rank. Therefore, we can guess a solution of the form

$$\mathbf{W}_{\pm,i} = \pm\mathbf{e}_i\mathbf{x}^\top \tag{40}$$

where the $\mathbf{e}_i$ are the $M$ coordinate vectors. This suggests that we have

$$\mathbf{w}_{\pm,i} = (\sqrt{\mathbf{x}\cdot\mathbf{x}})\mathbf{e}_i \tag{41}$$

This gives us our final $2M$ eigenmodes.

We can analyze the initial values of of the $\tilde{J}(\omega_i)$ as well. The components of the Jacobian can be written as:

$$(\mathbf{J}_v)_i \equiv \frac{\partial f(\mathbf{x})}{\partial\mathbf{v}_i} = \mathbf{U}_{im}\mathbf{x}_m \tag{42}$$

$$(\mathbf{J}_U)_{jm} \equiv \frac{\partial f(\mathbf{x})}{\partial\mathbf{U}_{jm}} = \mathbf{v}_j\mathbf{x}_m \tag{43}$$

From this form, we can deduce that $\mathbf{J}$ is orthogonal to the $0$ modes. We can also compute the conserved quantity. Let $J_+^2$ be the total weight in the positive eigenmodes, and $J_-^2$ be the total weight in the negative eigenmodes. A direct calculation shows that

$$\omega^{-1}(J_+^2 - J_-^2) = 2f(\mathbf{x}) \tag{44}$$

which implies that $E = 0$.

Therefore, the single-hidden layer linear model on one datapoint is equivalent to the quartic loss model with $E = 0$ and eigenvalues $\pm\sqrt{\mathbf{x}\cdot\mathbf{x}}$.

### A.3    CONNECTION TO BORDELON & PEHLEVAN (2022)

Since the one-hidden layer linear model has constant $\mathbf{Q}$, the models in Section F.1 of Bordelon & Pehlevan (2022) fall into the quadratic regression class. In the case of Section F.1.1, Equation 67, we can make the mapping to a $D = 1$ model explicit. The dynamics are equivalent to said model with a single eigenvalue $\omega_0$ if we make the identifications

$$\Delta = \tilde{z}, \; H_y = J_0^2, \; \gamma_0 = \sqrt{2\omega}, \; y = -E/2 \tag{45}$$

### A.4 Connection to NTH

The Neural Tangent Hierarchy (NTH) equations extend the NTK dynamics to account for changes in the tangent kernel by constructing an infinite sequence of higher order tensors which control the non-linear dynamics of learning Huang & Yau (2020). Truncation of the NTH equations at 3rd order is related to, but not the same as the quadratic regression model, as we will show here.

The 3rd order NTH equation describes the change in the tangent kernel $\mathbf{J}\mathbf{J}^\top$. Consider the $D \times D \times D$-dimensional kernel $\mathbf{K}_3$ whose elements are given by

$$(\mathbf{K}_3)_{\alpha\beta\gamma} = \frac{\partial^2 \mathbf{z}_\alpha}{\partial\boldsymbol{\theta}_i \partial\boldsymbol{\theta}_j} \mathbf{J}_{i\gamma}\mathbf{J}_{j\beta} + \frac{\partial^2 \mathbf{z}_\beta}{\partial\boldsymbol{\theta}_i \partial\boldsymbol{\theta}_j} \mathbf{J}_{i\gamma}\mathbf{J}_{j\alpha} \tag{46}$$

where repeated indices are summed over. In the NTH, for squared loss the change in the NTK $\mathbf{J}\mathbf{J}^\top$ is given by

$$\frac{d}{dt}\left(\mathbf{J}\mathbf{J}^\top\right)_{\alpha\beta} = -\eta(\mathbf{K}_3)_{\alpha\beta\gamma}\mathbf{z}_\gamma \tag{47}$$

For fixed $\mathbf{Q} = \frac{\partial^2 \mathbf{z}}{\partial\boldsymbol{\theta}\partial\boldsymbol{\theta}'}$, this equation is identical to the GF equations for the NTK in the quadratic regression model. We note that $\mathbf{K}_3$ is not constant under the quadratic regression model. Conversely, for fixed $\mathbf{K}_3$, $\frac{\partial^2 \mathbf{z}}{\partial\boldsymbol{\theta}\partial\boldsymbol{\theta}'}$ is not constant either. Therefore, the two methods can be used to construct different low-order expansions of the dynamics.

## B 2 Parameter Model

### B.1 Derivation of $\tilde{z}$-$T(0)$ equations

We can use the conserved quantity $E$ to write the dynamics in terms of $\tilde{z}$ and $T(0)$ only. Without loss of generality, let the eigenvalues are 1 and $\lambda$, with $-1 \leq \lambda \leq 1$. (We can achieve this by rescaling $\tilde{z}$.) Recall the dynamical equations

$$\tilde{z}_{t+1} - \tilde{z}_t = -\tilde{z}_t T_t(0) + \frac{1}{2}(\tilde{z}_t^2)T_t(1) \tag{48}$$

$$T_{t+1}(0) - T_t(0) = -\tilde{z}_t(2T_t(1) - \tilde{z}_t T_t(2)) \tag{49}$$

We will find substitutions for $T(1)$ and $T(2)$ in terms of $\tilde{z}$ and $T(0)$. Recall that we have

$$T(-1) = E + 2\tilde{z} \tag{50}$$

where $E$ is conserved throughout the dynamics (and indeed is a property of the landscape). We will use this definition to solve for $T(1)$ and $T(2)$.

Since $P = 2$, we can write $T(-1) = bT(0) + aT(1)$, for coefficients $a$ and $b$ which are valid for all combinations of $\tilde{J}$. If $\tilde{J}(\lambda) = 0$, we have $b = 1 - a$. If $\tilde{J}(1) = 0$, we have $1 = \lambda(1-a) + \lambda^2 a$. Solving, we have:

$$T(-1) = (1-a)T(0) + aT(1) \text{ for } a = -\frac{1}{\lambda} \tag{51}$$

The restrictions on $\lambda$ translate to $a \notin (-1, 1)$. In terms of the conserved quantity $E = T(-1) - 2\tilde{z}$, we have:

$$T(-1) = E + 2\tilde{z} \tag{52}$$

In order to convert the dynamics, we need to solve for $T(1)$ and $T(2)$ in terms of $T(0)$ and $\tilde{z}$. We have:

$$T(1) = \frac{1}{a}\left(T(-1) + (a-1)T(0)\right) = \frac{1}{a}\left(E + 2\tilde{z} + (a-1)T(0)\right) \tag{53}$$

We also have

$$T(2) = T(0) + \left(\frac{1-a}{a^2}\right)(T(0) - E - 2\tilde{z}) \tag{54}$$

This gives us

$$\tilde{z}_{t+1} - \tilde{z}_t = -\tilde{z}_t T_t(0) + \frac{1}{2a}(\tilde{z}_t^2)((a-1)T_t(0) + 2\tilde{z}_t + E) \tag{55}$$

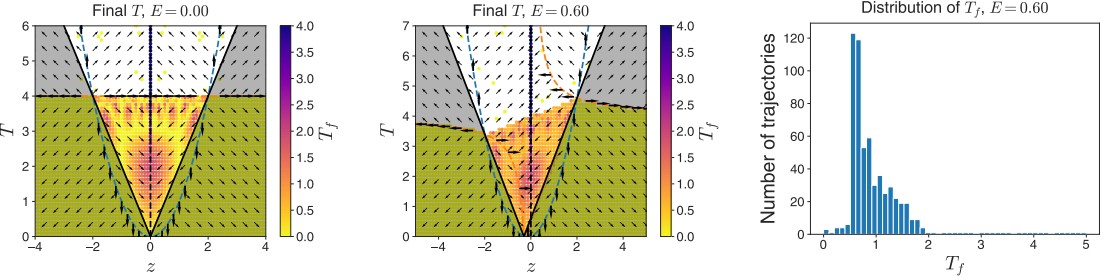

Figure 7: Phase portraits for symmetric model. Arrows indicate signs of changes in $\mathbf{z}$ and $T$, and grey area represents disallowed coordinates. Dynamics are run from an evenly spaced grid of initializations, and the final value of the curvature $T(0)$ is recorded. Nullclines representing $\tilde{z}_{t+1} - \tilde{z}_t = 0$ (blue) and $T_{t+1}(0) - T_t(0) = 0$ (orange) depend on $E$. Trajectories show progressive sharpening but no edge-of-stability effect (right).

$$T_{t+1}(0) - T_t(0) = -\frac{2}{a}\tilde{z}_t(2\tilde{z}_t + E + (a-1)T_t(0)) + z_t^2\left[T_t(0) + \left(\frac{1-a}{a^2}\right)(T_t(0) - E - 2\tilde{z}_t)\right] \tag{56}$$

If $\lambda = -\epsilon$ (that is, $a = \epsilon^{-1}$) we recover the equations from the main text.

The non-negativity of $\tilde{J}^2$ gives us constraints on the values of $\tilde{z}$ and $T$. For $a > 1$ (small negative second eigenvalue), the constraints are:

$$T > 2\tilde{z} + E, \ T > -(2\tilde{z} + E)/a \tag{57}$$

This is an upward-facing cone with vertex at $\tilde{z} = -E/2$ (Figure 8, left). For $a < -1$, the constraints are

$$-(2\tilde{z} + E)/a < T < 2\tilde{z} + E \tag{58}$$

This is a sideways facing cone with vertex at $\tilde{z} = -E/2$ (Figure 8, right). We see that in this case, there is a limited set of values of $T$ to converge to. Indeed, for $E = 0$, there is no convergence except at $T(0) = 0$. This why we focus on the case of one positive and one negative eigenvalue.

We can also solve for the nullclines - the curves where either $\tilde{z}_{t+1} - \tilde{z}_t = 0$ (blue in Figure 8), or $T_{t+1}(0) - T_t(0) = 0$ (orange in Figure 8). The nullcline $(\tilde{z}, f_{\tilde{z}}(\tilde{z}))$ for $\tilde{z}$ is given by

$$f_{\tilde{z}}(\tilde{z}) = \frac{\tilde{z}(2\tilde{z} + E)}{2a - (a-1)\tilde{z}} \tag{59}$$

The nullcline $(\tilde{z}, f_T(\tilde{z}))$ for $T(0)$ is given by

$$f_T(\tilde{z}) = -\frac{(a-1)\tilde{z} - 2a}{(a^2 - a + 1)\tilde{z} - 2a(a-1)}(2\tilde{z} + E) \tag{60}$$

The line $\tilde{z} = 0$ is also a nullcline.

For the symmetric model $\epsilon = 1$, the structure of the nullclines determines the presence or lack of progressive sharpening. For $E = 0$, there is no sharpening; the phase portrait (Figure 7, left) confirms this as the nullcline in $T_t(0)$ divides the space into two halves, one which converges, and the other which doesn't. However, when $E \neq 0$, the nullclines split, and there is a small region where progressive sharpening can occur (Figure 7, middle). However, there is still no edge-of-stability behavior in this case - there is no region where the trajectories cluster near $\lambda_{\max} = 2/\eta$ (Figure 7, right).

For the symmetric model $\epsilon = 1$, the structure of the nullclines determines the presence or lack of progressive sharpening. For $E = 0$, there is no sharpening; the phase portrait (Figure 7, left) confirms this as the nullcline in $T_t(0)$ divides the space into two halves, one which converges, and the other which doesn't. However, when $E \neq 0$, the nullclines split, and there is a small region where progressive sharpening can occur (Figure 7, middle). However, there is still no edge-of-stability behavior in this case - there is no region where the trajectories cluster near $\lambda_{\max} = 2/\eta$ (Figure 7, right).

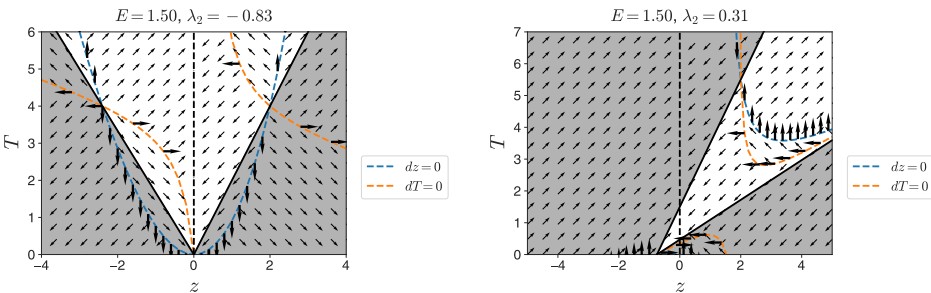

Figure 8: Phase planes of $D = 1$, $P = 2$ model. Grey region corresponds to parameters forbidden by positivity constraints on $\tilde{J}(\omega_i)^2$. For $\lambda > 0$, allowed region is smaller and intersects $\tilde{z} = 0$ at a small range only. Nullclines can be solved for analytically.

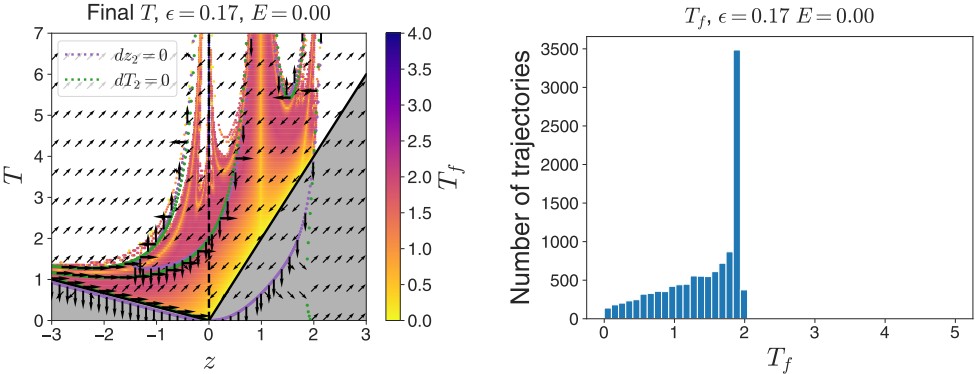

Figure 9: Phase portraits for $\epsilon = 0.17$, $E = 0$. Arrows indicate signs of changes in two step dynamics of $\tilde{z}$ and $T(0)$ (left), and grey area represents disallowed coordinates. Nullclines representing $\tilde{z}_{t+2} - \tilde{z}_t = 0$ (purple) and $T_{t+2}(0) - T_t(0) = 0$ (green) going in to $T(0) = 2$ almost overlap. Final $T(0)$ for initializations taken uniformly over $\tilde{z} - T(0)$ space show a peak near $T(0) = 2$ (right).

## B.2 TWO-STEP DYNAMICS

The two-step difference equations can be derived by iterating Equations 12 and 13. We have

$$\tilde{z}_{t+2} - \tilde{z}_t = p_0(\tilde{z}_t, \epsilon) + p_1(\tilde{z}_t, \epsilon)T_t(0) + p_2(\tilde{z}_t, \epsilon)T_t(0)^2 + p_3(\tilde{z}_t, \epsilon)T_t(0)^3 \quad (61)$$

$$T(0)_{t+2} - T_t(0) = q_0(\tilde{z}_t, \epsilon) + q_1(\tilde{z}_t, \epsilon)T_t(0) + q_2(\tilde{z}_t, \epsilon)T_t(0)^2 + q_3(\tilde{z}_t, \epsilon)T_t(0)^3 \quad (62)$$

Here the $p_i$ and $q_i$ are polynomials in $\tilde{z}$, maximum 9th order in $\tilde{z}$ and 6th order in $\epsilon$. They can be computed explicitly but we choose to omit the exact forms for now.

Numerical simulation of the dynamics for small $\epsilon$ reveals an edge of stability effect (Figure 9). We see that the distribution of final values of $T$ for random initializations has a peak near $T(0) = 2$ (right). By plotting the two-step dynamics, we can see that the two-step nullclines which go in to $T(0) = 2$ almost coincide (left). By studying these nullclines, we will be able to understand the edge of stability effect.

For fixed $\epsilon$, we can solve for the $\tilde{z}$ two-step nullclines ($\tilde{z}_{t+2} - \tilde{z}_t = 0$) and the $T$ nullclines ($T_{t+2}(0) - T_t(0) = 0$) using Cardano's formula to solve for $T$ as a function of $\tilde{z}$. In particular, each nullcline equation has a solution that goes through $\tilde{z} = 0$, $T(0) = 2$, independent of $\epsilon$. This is the family of solutions that we will focus on.

Let $(\tilde{z}, f_{\tilde{z}, \epsilon}(\tilde{z}))$ be the nullcline of $\tilde{z}$, and let $(\tilde{z}, f_{T, \epsilon}(\tilde{z}))$ be the nullcline of $T(0)$. We will show that the $T$ values of the nullclines, as a function of $\tilde{z}$ and $\epsilon$, is differentiable around $\tilde{z} = 0$, $\epsilon = 0$.

The nullclines are defined by the implicit equations

$$0 = 6\tilde{z}^3\epsilon - 2T\tilde{z} - 3T\tilde{z}^2(\epsilon - 1) - T\tilde{z}^3(\epsilon + 2)(2\epsilon + 1) + T^2\tilde{z} + \frac{7}{2}T^2\tilde{z}^2(\epsilon - 1)$$
$$+ \frac{1}{2}T^2\tilde{z}^3\left(9\epsilon^2 - 10\epsilon + 9\right) - \frac{1}{2}T^3\tilde{z}^2(\epsilon - 1) - \frac{1}{2}T^3\tilde{z}^3\left(3\epsilon^2 - 4\epsilon + 3\right) + O(\tilde{z}^4) \tag{63}$$

$$0 = -8\tilde{z}^2\epsilon - 12\tilde{z}^3(\epsilon - 1)\epsilon + 4T\tilde{z}(\epsilon - 1) + 2T\tilde{z}^2\left(3\epsilon^2 - \epsilon + 3\right) + 4T\tilde{z}^3(\epsilon - 1)\left(\epsilon^2 + 4\epsilon + 1\right)$$
$$- 2T^2\tilde{z}(\epsilon - 1) - T^2\tilde{z}^2\left(7\epsilon^2 - 8\epsilon + 7\right) - T^2\tilde{z}^3(\epsilon - 1)\left(9\epsilon^2 - \epsilon + 9\right) + T^3\tilde{z}^2\left(\epsilon^2 - \epsilon + 1\right)$$
$$+ T^3\tilde{z}^3(\epsilon - 1)\left(3\epsilon^2 - \epsilon + 3\right) + O(\tilde{z}^4) \tag{64}$$

We omit the higher order terms for now in anticipation of differentiating at $\tilde{z} = 0$ to use the implicit function theorem. Dividing by $\tilde{z}$, we have the equations

$$0 = 6\tilde{z}^2\epsilon - 2T - 3T\tilde{z}(\epsilon - 1) - T\tilde{z}^2(\epsilon + 2)(2\epsilon + 1) + T^2 + \frac{7}{2}T^2\tilde{z}(\epsilon - 1)$$
$$+ \frac{1}{2}T^2\tilde{z}^2\left(9\epsilon^2 - 10\epsilon + 9\right) - \frac{1}{2}T^3\tilde{z}(\epsilon - 1) - \frac{1}{2}T^3\tilde{z}^2\left(3\epsilon^2 - 4\epsilon + 3\right) + O(\tilde{z}^3) \tag{65}$$

$$0 = -8\tilde{z}\epsilon - 12\tilde{z}^2(\epsilon - 1)\epsilon + 4T(\epsilon - 1) + 2T\tilde{z}\left(3\epsilon^2 - \epsilon + 3\right) + 4T\tilde{z}^2(\epsilon - 1)\left(\epsilon^2 + 4\epsilon + 1\right)$$
$$- 2T^2(\epsilon - 1) - T^2\tilde{z}\left(7\epsilon^2 - 8\epsilon + 7\right) - T^2\tilde{z}^2(\epsilon - 1)\left(9\epsilon^2 - \epsilon + 9\right) + T^3\tilde{z}\left(\epsilon^2 - \epsilon + 1\right) \tag{66}$$
$$+ T^3\tilde{z}^2(\epsilon - 1)\left(3\epsilon^2 - \epsilon + 3\right) + O(\tilde{z}^3)$$

We immediately see that $\tilde{z} = 0, T = 2$ solves both equations for all $\epsilon$. Let $w(\epsilon, \tilde{z}, T)$ and $v(\epsilon, \tilde{z}, T)$ be the right hand sides of Equations 65 and 66 respectively. We have

$$\left.\frac{\partial w}{\partial T}\right|_{(0,0,2)} = 2, \quad \left.\frac{\partial v}{\partial T}\right|_{(0,0,2)} = 4 \tag{67}$$

In both cases the derivative is invertible. Therefore, $f_{\tilde{z},\epsilon}(\tilde{z})$ and $f_{T,\epsilon}(\tilde{z})$ are continuously differentiable in both $\tilde{z}$ and $\epsilon$ in some neighborhood of $0$. In fact, since $w$ and $v$ are analytic in all three arguments, $f_{\tilde{z},\epsilon}(\tilde{z})$ and $f_{T,\epsilon}(\tilde{z})$ are analytic as well.

We can use the analyticity to solve for the low-order structure of the nullclines. One way to compute the values of the derivatives is to define the nullclines as formal power series:

$$f_{\tilde{z}}(\tilde{z}) = 2 + \sum_{j=1}^{\infty}\sum_{k=1}^{\infty} a_{j,k}\epsilon^j\tilde{z}^k \tag{68}$$

$$f_T(\tilde{z}) = 2 + \sum_{j=1}^{\infty}\sum_{k=1}^{\infty} b_{j,k}\epsilon^j\tilde{z}^k \tag{69}$$

We can then solve for the first few terms of the series using Equations 65 and 66. From this procedure, we have:

$$f_{\tilde{z},\epsilon}(\tilde{z}) = 2 + 2\left(1 - \epsilon\right)\tilde{z} + 2\left(1 - \epsilon + \epsilon^2\right)\tilde{z}^2 + O(\tilde{z}^3) \tag{70}$$

$$f_{T,\epsilon}(\tilde{z}) = 2 - \frac{\left(2 - 3\epsilon + 2\epsilon^2\right)}{1 - \epsilon}\tilde{z} + \frac{1}{2}\left(4 - \epsilon + 4\epsilon^2\right)\tilde{z}^2 + O(\tilde{z}^3) \tag{71}$$

The difference $f_{\Delta,\epsilon}(\tilde{z})$ between the two is:

$$f_\Delta(\tilde{z}) \equiv f_{\tilde{z}}(\tilde{z}) - f_T(\tilde{z}) = -\frac{\epsilon}{1 - \epsilon}\tilde{z} - \frac{3}{2}\epsilon\tilde{z}^2 + O(\tilde{z}^3) \tag{72}$$

As $\epsilon$ decreases, for the low order terms the distance between the nullclines also decreases.

We can show that the difference goes as $\epsilon$. The one-step dynamical equations for $\epsilon = 0$ are

$$\tilde{z}_{t+1} - \tilde{z}_t = -\tilde{z}_t T_t(0) + \frac{1}{2}\tilde{z}_t^2 T_t(0) \tag{73}$$

$$T_{t+1}(0) - T_t(0) = -2\tilde{z}_t T_t(0) + z_t^2 T_t(0) \tag{74}$$

Therefore, $\Delta\tilde{z} = 2\Delta T$. This means that both the one step AND two-step nullclines are identical. Since $f_{\tilde{z},0}(\tilde{z}) = f_{T,0}(\tilde{z})$, and both are differentiable with respect to $\epsilon$, we have:

$$f_{\tilde{z},\epsilon}(\tilde{z}) - f_{T,\epsilon}(\tilde{z}) = \epsilon f_{\Delta,\epsilon}(\tilde{z}) \tag{75}$$

for some function $f_{\Delta,\epsilon}(\tilde{z})$ which is analytic in $\epsilon$ and $\tilde{z}$ in a neighborhood around $(0, 0)$.

### B.3 Two-step dynamics of $y$

It is useful to define dynamical equations in coordinates $(\tilde{z}, y)$ where $y$ is the difference between $T(0)$ and the $\tilde{z}$ nullcline:

$$y \equiv T(0) - f_{\tilde{z}, \epsilon}(\tilde{z}) \tag{76}$$

To lowest order in $\tilde{z}$ and $\epsilon$ we have

$$y = T(0) - 2 - 2(1 - \epsilon)\tilde{z} - 2(1 - \epsilon + \epsilon^2)\tilde{z}^2 + O(\tilde{z}^3) \tag{77}$$

We note that $y = 0$, at $\tilde{z} = 0$ corresponds to $T(0) = 2$. $y$ near but slightly less than $0$ is equivalent to edge-of-stability behavior. For positive $\tilde{z}$, $y = 0$ implies $T(0) > 2$.

We can write the dynamics of $\tilde{z}$ and $y$. The dynamics for $\tilde{z}$ are given by:

$$\tilde{z}_{t+2} - \tilde{z}_t = p_0(\tilde{z}_t, \epsilon) + p_1(\tilde{z}_t, \epsilon)(y_t + f_{\tilde{z}, \epsilon}(\tilde{z}_t)) + p_2(\tilde{z}_t, \epsilon)(y_t + f_{\tilde{z}, \epsilon}(\tilde{z}_t))^2 + p_3(\tilde{z}_t, \epsilon)(y_t + f_{\tilde{z}, \epsilon}(\tilde{z}_t))^3 \tag{78}$$

We know that the right hand side of this equation is analytic in $\tilde{z}$, $\epsilon$, and (trivially) $y$ as well. By evaluating the multiple continuous derivatives of $f$, we can write:

$$\tilde{z}_{t+2} - \tilde{z}_t = 2y_t \tilde{z}_t + y_t^2 \tilde{z}_t f_{1,\epsilon}(\tilde{z}_t, y_t) + y_t \tilde{z}_t^2 f_{2,\epsilon}(\tilde{z}_t) \tag{79}$$

Here, $f_{1,\epsilon}$ and $f_{2,\epsilon}$ are analytic in $\tilde{z}$, $\epsilon$, and $y$ in some neighborhood around $0$.

This means that we have the bounds

$$|f_{1,\epsilon}(\tilde{z}, y)| < F_1, \quad |f_{2,\epsilon}(\tilde{z}, y)| < F_2 \tag{80}$$

for $(\tilde{z}, \epsilon, y) \in [-\tilde{z}_d, \tilde{z}_d] \times [0, \epsilon_d] \times [-y_d, y_d]$ for some non-negative constants $F_1$ and $F_2$. Note that this bound is independent of $\epsilon$.

Now we consider the dynamics of $y$. We have:

$$y_{t+2} - y_t = T_{t+2}(0) - T_t(0) - f_{\tilde{z}, \epsilon}(\tilde{z}_{t+2}) + f_{\tilde{z}, \epsilon}(\tilde{z}_t) \tag{81}$$

Since $\lim_{\tilde{z} \to 0, y \to 0} \tilde{z}_{t+2} = 0$, $f_{\tilde{z}, \epsilon}(\tilde{z}_{t+2})$ is analytic in some neighborhood of $(0, 0, 0)$. Therefore $y_{t+2} - y_t$ is analytic as well. Substituting, we have

$$\begin{aligned} y_{t+2} - y_t = &\ q_0(\tilde{z}_t, \epsilon) + q_1(\tilde{z}_t, \epsilon)[y + f_{\tilde{z}, \epsilon}(\tilde{z})] + q_2(\tilde{z}_t, \epsilon)[y + f_{\tilde{z}, \epsilon}(\tilde{z})]^2 + q_3(\tilde{z}_t, \epsilon)[y + f_{\tilde{z}, \epsilon}(\tilde{z})]^3 \\ &- f_{\tilde{z}, \epsilon}(\tilde{z}_t + 2y_t \tilde{z}_t + y_t^2 \tilde{z}_t f_{1,\epsilon}(\tilde{z}_t, y_t) + y_t \tilde{z}_t^2 f_{2,\epsilon}(\tilde{z}_t)) + f_{\tilde{z}, \epsilon}(\tilde{z}_t) \end{aligned} \tag{82}$$

If we write $f_{\tilde{z}, \epsilon}(\tilde{z}) = f_{T, \epsilon}(\tilde{z}) + \epsilon f_{\Delta, \epsilon}(\tilde{z})$, then we can write:

$$\begin{aligned} y_{t+2} - y_t = &\ q_0(\tilde{z}_t, \epsilon) + q_1(\tilde{z}_t, \epsilon)[f_{T, \epsilon}(\tilde{z})] + q_2(\tilde{z}_t, \epsilon)[f_{T, \epsilon}(\tilde{z})]^2 + q_3(\tilde{z}_t, \epsilon)[f_{T, \epsilon}(\tilde{z})]^3 \\ &\ 2q_2(\tilde{z}_t, \epsilon)[f_{T, \epsilon}(\tilde{z})(y + \epsilon f_{\Delta, \epsilon}(\tilde{z}))] + 3q_3(\tilde{z}_t, \epsilon)[(f_{T, \epsilon}(\tilde{z}))(y + \epsilon f_{\Delta, \epsilon}(\tilde{z}))^2 + (f_{T, \epsilon}(\tilde{z}))^2(y + \epsilon f_{\Delta, \epsilon}(\tilde{z}))] \\ &\ q_0(\tilde{z}_t, \epsilon) + q_1(\tilde{z}_t, \epsilon)[y + \epsilon f_{\Delta, \epsilon}(\tilde{z})] + q_2(\tilde{z}_t, \epsilon)[y + \epsilon f_{\Delta, \epsilon}(\tilde{z})]^2 + q_3(\tilde{z}_t, \epsilon)[y + \epsilon f_{\Delta, \epsilon}(\tilde{z})]^3 \\ &- f_{\tilde{z}, \epsilon}(\tilde{z}_t + 2y_t \tilde{z}_t + y_t^2 \tilde{z}_t f_{1,\epsilon}(\tilde{z}_t, y_t) + y_t \tilde{z}_t^2 f_{2,\epsilon}(\tilde{z}_t)) + f_{\tilde{z}, \epsilon}(\tilde{z}_t) \end{aligned} \tag{83}$$

By the definition of the nullclines, the first four terms vanish. Once again using the differentiability of the nullclines, as well as $f_{1,\epsilon}$ and $f_{2,\epsilon}$, we can rewrite the dynamics in terms of the expansion:

$$y_{t+2} - y_t = -2(4 - 3\epsilon + 4\epsilon^2)y_t \tilde{z}_t^2 - 4\epsilon \tilde{z}_t^2 + y_t^2 \tilde{z}_t^2 g_{1,\epsilon}(\tilde{z}_t, y_t) + \epsilon \tilde{z}_t^3 g_{2,\epsilon}(\tilde{z}_t) \tag{84}$$

Here $g_{1,\epsilon}$ and $g_{2,\epsilon}$ are analytic near zero in $\tilde{z}$, $y$, and $\epsilon$. We have the bounds

$$|g_{1,\epsilon}(\tilde{z}, y)| < G_1, \quad |g_{1,\epsilon}(\tilde{z}, y)| < G_2 \tag{85}$$

for $(\tilde{z}, \epsilon, y) \in [-\tilde{z}_d, \tilde{z}_d] \times [0, \epsilon_d] \times [-y_d, y_d]$ for some non-negative constants $G_1$ and $G_2$. This bound is also independent of $\epsilon$.

We can summarize these bounds in the following lemma:

**Lemma B.1.** *Define $y = T - f_{\tilde{z}}(\tilde{z})$. The two step dynamics of $\tilde{z}$ and $y$ are given by*

$$\tilde{z}_{t+2} - \tilde{z}_t = 2y_t\tilde{z}_t + y_t^2\tilde{z}_t f_{1,\epsilon}(\tilde{z}_t, y_t) + y_t\tilde{z}_t^2 f_{2,\epsilon}(\tilde{z}_t) \tag{86}$$

$$y_{t+2} - y_t = -2(4 - 3\epsilon + 4\epsilon^2)y_t\tilde{z}_t^2 - 4\epsilon\tilde{z}_t^2 + y_t^2\tilde{z}_t^2 g_{1,\epsilon}(\tilde{z}_t, y_t) + \epsilon\tilde{z}_t^3 g_{2,\epsilon}(\tilde{z}_t, y_t) \tag{87}$$

*Where $f_{1,\epsilon}$, $f_{2,\epsilon}$, $g_{1,\epsilon}$, $g_{2,\epsilon}$ are all analytic in $\tilde{z}$, $y$, and $\epsilon$. Additionally, there exist positive $\tilde{z}_c$, $y_c$, and $\epsilon_c$ such that*

$$|f_{1,\epsilon}(\tilde{z}, y)| < F_1, \; |f_{2,\epsilon}(\tilde{z}, y)| < F_2, \; |g_{1,\epsilon}(\tilde{z}, y)| < G_1, \; |g_{1,\epsilon}(\tilde{z}, y)| < G_2 \tag{88}$$

*for all $(\tilde{z}, \epsilon, y) \in [-\tilde{z}_d, \tilde{z}_d] \times [0, \epsilon_d] \times [-y_d, y_d]$, where $F_1$, $F_2$, $G_1$, and $G_2$ are all non-negative constants.*

We can use this Lemma to analyze the dynamics for small fixed $\epsilon$, for small initializations of $\tilde{z}$, $y$. The control of the higher order terms will allow for an analysis which focuses on the effects of the lower order terms.

## B.4 PROOF OF THEOREM 2.1

Using Lemma B.1, the dynamics in $\tilde{z}$ and $y$ can be written as:

$$\tilde{z}_{t+2} - \tilde{z}_t = 2y_t\tilde{z}_t + y_t^2\tilde{z}_t f_{1,\epsilon}(\tilde{z}_t, y_t) + y_t\tilde{z}_t^2 f_{2,\epsilon}(\tilde{z}_t) \tag{89}$$

$$y_{t+2} - y_t = -2(4 - 3\epsilon + 4\epsilon^2)y_t\tilde{z}_t^2 - 4\epsilon\tilde{z}_t^2 + y_t^2\tilde{z}_t^2 g_{1,\epsilon}(\tilde{z}_t, y_t) + \epsilon\tilde{z}_t^3 g_{2,\epsilon}(\tilde{z}_t, y_t) \tag{90}$$

Let $\epsilon < \epsilon_d$. Then we can use the bounds from Lemma B.1 to control the contributions of the higher order terms to the dynamics:

**Lemma B.2.** *Given constants $A > 0$ and $B > 0$, there exist $\tilde{z}_c$ and $y_c$ such that for $\tilde{z} \in [0, 2\tilde{z}_c]$, $y \in [-y_c, y_c]$, we have the bounds:*

$$|y^2\tilde{z}f_{1,\epsilon}(\tilde{z}, y) + y\tilde{z}^2 f_{2,\epsilon}(\tilde{z})| \le A|2y\tilde{z}| \tag{91}$$

$$|y^2\tilde{z}^2 g_{1,\epsilon}(\tilde{z}, y)| \le \frac{B}{8}|2(4 - 3\epsilon + 4\epsilon^2)y\tilde{z}^2| \tag{92}$$

$$|\epsilon\tilde{z}^3 g_{2,\epsilon}(\tilde{z}, y)| \le \frac{B}{4}|4\epsilon\tilde{z}^2| \tag{93}$$

*Proof.* We begin by the following decomposition:

$$|y^2\tilde{z}f_{1,\epsilon}(\tilde{z}, y) + y\tilde{z}^2 f_{2,\epsilon}(\tilde{z})| \le |y^2\tilde{z}f_{1,\epsilon}(\tilde{z}, y)| + |y\tilde{z}^2 f_{2,\epsilon}(\tilde{z})| \tag{94}$$

From Lemma B.1, there exists a region $[-\tilde{z}_d, \tilde{z}_d] \times [0, \epsilon_d] \times [-y_d, y_d]$ where the magnitudes of $f_{1,\epsilon}$, $f_{2,\epsilon}$, $g_{1,\epsilon}$, and $g_{2,\epsilon}$ are bounded by $F_1$, $F_2$, $G_1$, and $G_2$ respectively.

$$|y^2\tilde{z}f_1(\tilde{z}, y) + y\tilde{z}^2 f_2(\tilde{z})| \le F_1 y^2\tilde{z} + F_2 y\tilde{z}^2 \tag{95}$$

$$|y^2\tilde{z}^2 g_1(\tilde{z}, y)| \le G_1 y^2\tilde{z}^2 \tag{96}$$

$$|\tilde{z}^3 g_2(\tilde{z}, y)| \le G_2\tilde{z}^3 \tag{97}$$

Define $\tilde{z}_c$ and $y_c$ as

$$y_c = \min(A/F_1, B/2G_1, y_d), \; \tilde{z}_c = \min(A/2F_2, B/2G_2, y_c) \tag{98}$$

The desired bounds follow immediately. $\square$

We consider an initialization $(\tilde{z}_0, y_0)$ such that $\tilde{z}_0 \le \tilde{z}_c$ and $y_0 \le y_c$, and $y_0 \le \tilde{z}_0^2$. Armed with Lemma B.2, we can analyze the dynamics. There are two phases; in the first phase, $\tilde{z}$ is increasing, and $y$ is decreasing. The first phase ends when $y$ becomes negative for the first time - reaching a value of $O(\epsilon)$. In the second phase, $\tilde{z}$ is decreasing, and $y$ stays negative and $O(\epsilon)$.

### B.4.1   PHASE ONE

Let $t_{sm}$ be the time such that for $t \leq t_{sm}$, $\tilde{z}_t \leq 2\tilde{z}_0$. (We will later show that $\tilde{z}_t \leq 2\tilde{z}_0$ over the whole dynamics.) For $t \leq t_{sm}$, using Lemma B.2, the change in $\tilde{z}$ can be bounded from below by

$$\tilde{z}_{t+2} - \tilde{z}_t \geq 2y_t\tilde{z}_t(1 - A) \tag{99}$$

Therefore at initialization, $\tilde{z}$ is increasing. It remains increasing until $y_t$ becomes negative, or $\tilde{z}_t \geq 2\tilde{z}_0$. We want to show that $y_t$ becomes negative before $\tilde{z}_t \geq 2\tilde{z}_0$.

For any $t \leq t_{sm}$, Lemma B.2 gives the following upper bound on $y_{t+2} - y_t$:

$$y_{t+2} - y_t \leq -(8 - B)y_t\tilde{z}_t^2 - (4 - B)\epsilon\tilde{z}_t^2 \tag{100}$$

Let $t_-$ be the first time that $y_t$ becomes negative. Since $\tilde{z}_t$ is increasing for $t \leq t_-$, we have

$$y_{t+2} - y_t \leq -(8 - B)y_t\tilde{z}_0^2 - (4 - B)\epsilon\tilde{z}_0^2 \tag{101}$$

This gives us the following bound on $y_t$:

$$y_t \leq y_0 e^{-(8-B)\tilde{z}_0^2 t} \tag{102}$$

valid for $t \leq t_-$ and $t \leq t_{sm}$.

We will now show that $t_- < t_{sm}$. Suppose that $t_{sm} \leq t_-$. Then at $t_{sm} + 2$, $\tilde{z}_{t_{sm}+2} > 2\tilde{z}_0$ for the first time. Summing the bound in Equation 99, we have:

$$\tilde{z}_{t_{sm}+2} - \tilde{z}_0 \leq \sum_{t=0}^{t_{sm}} 2y_t\tilde{z}_t(1 + A) \leq 4\tilde{z}_0(1 + A)\sum_{t=0}^{t_{sm}} y_t \tag{103}$$

where the second bound comes from the definition of $t_{sm}$. Using our bound on $y_t$, we have:

$$\tilde{z}_{t_{sm}+2} - \tilde{z}_0 \leq 4\tilde{z}_0(1 + A)\sum_{s=0}^{t_{sm}} y_0 e^{-(8-B)\tilde{z}_0^2 s} \leq \frac{(1 + A)}{2}\frac{y_0}{\tilde{z}_0} \tag{104}$$

Since $y_0 \leq \tilde{z}_0^2$, $\tilde{z}_{t_{sm}+2} \leq 2\tilde{z}_0$. However, by assumption $\tilde{z}_{t_{sm}+2} > 2\tilde{z}_0$. We arrive at a contradiction; $t_{sm}$ is not less than or equal to $t_-$.

There are three possibilities: the first is that $t_-$ is well-defined, and $t_- < t_{sm}$. Another possibility is that $t_-$ is not well-defined - that is, $y_t$ never becomes negative. In this case the bounds we derived are valid for all $t$. Therefore using Equation 102, there exists some time $t_\epsilon$ where $y_{t_\epsilon} < (4 - B)\epsilon\tilde{z}_0^2$. Then, using Equation 101 we have $y_{t_\epsilon+2} < 0$. Therefore, we conclude that $t_-$ is finite and less than $t_{sm}$.

Since the well defined value $t_- < t_{sm}$, when $y$ first becomes negative, $\tilde{z}_{t_-} \leq 2\tilde{z}_0$. This means that we can continue to apply the bounds from Lemma B.2 at the start of the next phase. At $t = t_- - 2$, applying Lemma B.2 and $\tilde{z}_{t_-} \leq 2\tilde{z}_0$, we have

$$y_{t_-} - y_{t_--2} \geq -4(8 + B)y_{t_--2}\tilde{z}_0^2 - 4(4 + B)\epsilon\tilde{z}_0^2 \tag{105}$$

which gives us $y_{t_-} \geq -4(4 + B)\epsilon\tilde{z}_0^2$. This concludes the first phase. To summarize we have

$$-4(4 + B)\epsilon\tilde{z}_0^2 < y_{t_-} \leq 0, \; \tilde{z}_{t_-} \leq 2\tilde{z}_0 \tag{106}$$

### B.4.2   PHASE TWO

Now consider the second phase of the dynamics. We will show that $y$ remains negative and $O(\epsilon)$, and $\tilde{z}$ decreases to 0. While $y$ is negative, $\tilde{z}$ decreases. While $y \geq -y_0$, from Lemma B.2 we have

$$\tilde{z}_{t+2} - \tilde{z}_t \leq (1 - A)2y_t\tilde{z}_t \tag{107}$$

Therefore as long as $-y_0 \leq y < 0$, $\tilde{z}_t$ is decreasing. If this is true for all subsequent $t$, $\tilde{z}_0$ will converge to 0.

We will now show that $y$ remains negative and $O(\epsilon)$, concluding the proof. Let $y^* = -\frac{\epsilon}{2-(3/2)\epsilon+2\epsilon^2}$. We can re-write the dynamical equation for $y$ as

$$y_{t+2} - y_t = -2(4 - 3\epsilon + 4\epsilon^2)\tilde{z}_t^2(y_t - y^*) + y_t^2\tilde{z}_t^2 g_1(\tilde{z}_t, y_t) + \tilde{z}_t^3 g_2(\tilde{z}_t, y_t) \tag{108}$$

Applying Lemma B.2 to the higher order terms, we have:

$$y_{t+2} - y_t \leq -2(4 - 3\epsilon + 4\epsilon^2)\tilde{z}_t^2(y_t - y^*) + B(|y_t| + \epsilon)\tilde{z}_t^2 \tag{109}$$

$$y_{t+2} - y_t \geq -2(4 - 3\epsilon + 4\epsilon^2)\tilde{z}_t^2(y_t - y^*) - B(|y_t| + \epsilon)\tilde{z}_t^2 \tag{110}$$

These inequalities are valid as long as $|y_t| < y_c$.

At $t_-$, $y^* < y_t < 0$. When $y^* < y_t < 0$, then $|y_t| \leq |y^*|$. Note that $\epsilon < 2|y^*|$. From Equation 109, we have

$$y_{t+2} - y_t \leq -2(4 - 3\epsilon + 4\epsilon^2)\tilde{z}_t^2(y_t - y^*) + B(-y_t + \epsilon)\tilde{z}_t^2 \tag{111}$$

From this inequality we can conclude that

$$y_{t+2} \leq (1 - 2(4 - 3\epsilon + 4\epsilon^2)\tilde{z}_t^2 - B)y_t + \tilde{z}_t^2[2(4 - 3\epsilon + 4\epsilon^2)y^* + B\epsilon] \tag{112}$$

If $B < 1$, then both terms are negative. We can conclude that if $y^* < y_t < 0$, $y_{t+2} < 0$. In fact, from the last term we can conclude that $y_{t+2} < -4\epsilon\tilde{z}_t^2$.

Now we must show that when $y^* < y_t < 0$, $y_{t+2}$ does not become too negative (namely, smaller than $-y_c$). Using Equation 110, we have:

$$y_{t+2} > y^*(1 + 3B\tilde{z}_0^2) \text{ if } y_t > y^* \tag{113}$$

This means that if $y_t$ starts larger than $y^*$, it will be at most $3B\tilde{z}_0^2 y^*$ below $y^*$ at the next step. Since $B < 1$, $y_{t+2} > -y_c$ if $y^* < y_t < 0$.

Finally, we will show that if $y^*(1 + 3B/(8 - B)) < y_t < y^*$, $y^*(1 + 3B/(8 - B)) < y_{t+2} < 0$. Since $y_{t_-+2}$ fits this condition, we can conclude that $y_t$ is negative for all $t > t_-$, with magnitude bounded from below by $y^*(1 + 3B/(8 - B))$, and complete the proof.

We will first show that $y^*(1 + 3B/(8 - B)) < y_t$ implies that $y^*(1 + 3B/(8 - B)) < y_{t+2}$. Let $y_t = (1 + \delta_t)y^*$, for $\delta_t < 3B/(8 - B)$. We will show that $\delta_{t+2} < 3B/(8 - B)$. Using Equation 110, we have:

$$y_{t+2} \geq (1 + \delta_t)y^* - 8\tilde{z}_t^2\delta_t y^* - B\tilde{z}_t^2(\epsilon - (1 + \delta_t)y^*) \tag{114}$$

Substituting $y_{t+2} = (1 + \delta_{t+2})y^*$, and dividing both sides by $y^*$ we have

$$\delta_{t+2} - \delta_t \leq -(8 - B)\tilde{z}_t^2\delta_t + 3B\tilde{z}_t^2 \tag{115}$$

If $\delta_t < 3B/(8 - B)$, then we have $\delta_{t+2} < 3B/(8 - B)$ as desired.

Finally, we will show that $0 < \delta_t < 3B/(8 - B)$ implies that $\delta_{t+2} > -1$ - that is, $[1 + 3B/(8 - B)]y^* < y_t < y^*$ implies $[1 + 3B/(8 - B)]y^* < y_{t+2} < 0$. Equation 109 implies

$$y_{t+2} \leq (1 + \delta_t)y^* - 8\tilde{z}_t^2\delta_t y^* + B\tilde{z}_t^2(\epsilon - (1 + \delta_t)y^*) \tag{116}$$

which gives us

$$\delta_{t+2} - \delta_t \geq -(8 - B)\tilde{z}_t^2\delta_t - 3B\tilde{z}_t^2 \tag{117}$$

If $\delta_t > 0$ implies

$$\delta_{t+2} > -3B\tilde{z}_t^2 \tag{118}$$

If $3B\tilde{z}_0^2 < 1$, then $\delta_{t+2} > -1$. This means that $y_{t+2} < 0$ if $[1 + 3B/(8 - B)]y^* < y_t < y^*$.

Finally, we make some choices of $B$ and $\tilde{z}_0$ to guarantee convergence. Choose $\tilde{z}_0^2 < 3/7$, and choose $B < \frac{1}{2}$. Then in summary, what we have shown for phase two is:

- At the start of the phase (time $t_-$), $y^* < y_{t_-} < 0$.
- If $y^* < y_t < 0$, $t > t_-$, $y^*(1 + 3B\tilde{z}_0^2) < y_{t+2} < -4\epsilon\tilde{z}_t^2$.
- If $[1 + 3B/(8 - B)]y^* < y_t < y^*$, $t > t_-$, $[1 + 3B/(8 - B)]y^* < y_{t+2} < 0$.

Through our choices of $\tilde{z}_0$ and $B$, we know that $[1 + 3B/(8 - B)]y^* < y^*(1 + 3B\tilde{z}_0^2)$. Therefore, the entire trajectory for $t > t_-$ is accounted for by these regions, and $[1 + 3B/(8 - B)]y^* < y_t < 0$ for all $t > t_-$. Additionally, we know that at least once every 2 steps, $y_t < -4\epsilon\tilde{z}_t^2$. This means that the dynamics of $\tilde{z}_t$ can be bounded from above by

$$\tilde{z}_{t+2} - \tilde{z}_t \leq -2\epsilon^2\tilde{z}_t^4 \tag{119}$$

From this we can conclude that $\tilde{z}_t$ converges to 0.

Therefore, for any positive initialization with $\tilde{z}_0 \leq \tilde{z}_c$, $y_0 \leq y_c$, and $y_0 \leq \tilde{z}_0^2$, we have:

$$\lim_{t\to\infty} \tilde{z}_t \to 0, \quad \lim_{t\to\infty} y = -y_f \tag{120}$$

where $y_f = O(\epsilon)$.

Now we can prove the statement of Theorem 2.1. Given a model with $\epsilon \leq \epsilon_c$, there is a continuous mapping between $\boldsymbol{\theta} - \eta$ space and $\tilde{z} - y$ space. Since the region $[0, \tilde{z}_c) \times [0, y_c)$ in $\tilde{z} - y$ space displays edge-of-stability behavior for $\epsilon$-independent $\tilde{z}_c$ and $y_c$ (that is, $T_t(0)$ converges to within $O(\epsilon)$ of 2 in that region), the inverse image of that neighborhood is a neighborhood in $\boldsymbol{\theta} - \eta$ space that displays edge-of-stability behavior. This concludes the proof.

## B.5   LOW ORDER DYNAMICS

In order to predict the final value of $y$, and understand the convergence to the fixed point, We can study the low order dynamics in $\tilde{z}$ and $y$. The low order dynamical equations are:

$$\tilde{z}_{t+2} - \tilde{z}_t = 2y_t\tilde{z}_t \tag{121}$$

$$y_{t+2} - y_t = -2(4 - 3\epsilon + 4\epsilon^2)y_t\tilde{z}_t^2 - 4\epsilon\tilde{z}_t^2 \tag{122}$$

For these reduced dynamics, we can show the following:

**Theorem B.3.** *For the dynamics defined by Equations 121 and 122, for $\epsilon \ll 1$, for positive initializations $\tilde{z}_0 \ll 1$, $y_0 \ll 1$ with the additional constraints $-\epsilon \log(\epsilon) \ll 16\tilde{z}_0^2$ and $y_0 < 2\tilde{z}_0^2$, we have*

$$\lim_{t\to\infty} \tilde{z}_t = 0, \quad \lim_{t\to\infty} y_t = -\epsilon/2 + O(\epsilon^2) \tag{123}$$

*Proof.* The proof distinguishes two phases in the time evolution:

- Phase 1: $\tilde{z}$ starts positive and increases, $y$ starts positive and decreases. At the end of the phase we want $\tilde{z}_t \leq 2\tilde{z}_0$ and $y$ to be negative but bounded by $-16\tilde{z}_0^2\epsilon$.

- Phase 2: $\tilde{z}$ decreases slowly, and $y$ settles to the fixed point (relatively) quickly, up to error $O(\epsilon^2)$.

Let $\epsilon \ll 1$. Consider an initialization $(\tilde{z}_0, y_0)$ where both variables are positive, such that $\tilde{z}_0 \ll 1$, $\epsilon \log(\epsilon) \ll \tilde{z}_0^2$, and $y_0 \ll \tilde{z}_0^2$. From Equations 121 and 122, we see that the dynamics of $y$ will depend on the balance of the two terms.

Initially $\tilde{z}$ increases and $y$ decreases. We analyze the dynamics of $y$ assuming that $\tilde{z}$ is fixed, and then compute the corrections.

Phase 1. At initialization, the first term in the dynamics dominates, since by assumption $\epsilon\tilde{z}_t^2 \ll y_t\tilde{z}_t^2 \ll$. Since $\tilde{z}_0^2 \ll 1$, $y$ initially decreases exponentially with decay rate bounded from above by $8\tilde{z}_0^2$. Therefore within $\log(-\epsilon/y_0)/8\tilde{z}_0^2$ steps, $y < \epsilon$.

At this point, the rate of change of $y$ is at least $-4\epsilon\tilde{z}_0^2$. Therefore, in no more than $1/4\tilde{z}_0^2$ additional steps, $y$ becomes negative. Let $t_-$ be the first time that $y$ becomes negative. We note that $y_{t_-} \geq -4\epsilon\tilde{z}_0^2$ under this analysis - the first term in Equation 122 is less than $y_t$ in magnitude, so the smallest value that $y_{t+2}$ can take if $y_t$ is positive is $-4\epsilon\tilde{z}_0^2$.

We can now understand the corrections due to the change in $\tilde{z}$. We note that $e^{-8\tilde{z}_0^2 t}$ is an upper bound for $y$ - since $\tilde{z}$ is increasing, and the $-4\epsilon\tilde{z}_t^2$ decreases $y$ faster than exponential decay from the first term. Since $\tilde{z}$ is increasing, $y_t \geq e^{-8\tilde{z}_0^2 t}$ as long as $y$ remains positive ($t < t_-$). Let $t_{sm}$ be a time such that $\tilde{z}_{t_{sm}} < 2\tilde{z}_0$. We can bound the change in $\tilde{z}_t$ for $t < t_{sm}$. We know that $y_t \geq y_0 e^{-8\tilde{z}_0^2 t}$. The change in $\tilde{z}$ can be bounded by

$$\tilde{z}_{t_{sm}} - \tilde{z}_0 \leq \sum_{t=0}^{t_{sm}} 2z_t y_t \leq 4\tilde{z}_0 \sum_{t=0}^{t_{sm}} y_t \leq 4\tilde{z}_0 y_0 \sum_{t=0}^{t_{sm}} e^{-8\tilde{z}_0^2 t} \leq \frac{1}{2} \cdot \frac{y_0}{\tilde{z}_0}. \tag{124}$$

If $y_0 < 2\tilde{z}_0^2$, then the bound holds independent of the value of $t_{sm}$, as long as the bound on $y$ is correct. We know that the bound on $y$ is correct until time $t_-$; therefore, $t_{sm} \geq t_-$.

Phase 2. This proves that there exists a time $t_-$, such that $\tilde{z}_{t_-} \leq 2\tilde{z}$, and $-16\tilde{z}_0^2\epsilon \leq y_{t_-} \leq 0$. Now that $y$ is negative, it will stay negative, and $\tilde{z}$ will decrease until it reaches $0$. In order to understand the dynamics, we will use a change of coordinates. Consider solving Equation 122 for $y_{t+2} - y_t = 0$ for $\tilde{z}_t \neq 0$. We have

$$y^* = -\frac{\epsilon}{2 - 3/2\epsilon + 2\epsilon^2} \tag{125}$$

Consider now the coordinate $\delta_t$ defined by the equation

$$y_t = -(1 + \delta_t)\frac{\epsilon}{2 - 3/2\epsilon + 2\epsilon^2} \tag{126}$$

The dynamics of $\delta_t$ are given by

$$\delta_{t+2} = (1 - 2(4 - 3\epsilon + 4\epsilon^2)\tilde{z}_t^2)\delta_t \tag{127}$$

Since $\tilde{z}_t \ll 1$, $\delta_t$ is strictly decreasing in magnitude. We can bound $\delta_t$ from above by

$$|\delta_t| \leq \exp\left(-8 \sum_{s=t_-}^{t} \tilde{z}_s^2\right) |\delta_{t_-}| \tag{128}$$

Since $\delta$ starts negative, and is decreasing in magnitude, we know that $y_t > -\frac{\epsilon}{2 - 3/2\epsilon + 2\epsilon^2}$. This means that we can bound $\tilde{z}_t$ by

$$\tilde{z}_t \geq 2e^{-\epsilon t}\tilde{z}_0 \tag{129}$$

Substitution gives us the following bound on $\delta_t$:

$$|\delta_t| \leq \exp\left(-8 \sum_{s=t_-}^{t} 4e^{-2\epsilon s}\tilde{z}_0^2\right) |\delta_{t_-}| \tag{130}$$

Using the integral approximation for the sum, the bound becomes

$$|\delta_t| \leq \exp\left(-32\tilde{z}_0^2 \int_0^t e^{-2\epsilon s}ds\right) \delta_{t_-} = \exp\left(-16\tilde{z}_0^2/\epsilon(1 - e^{-2\epsilon t})\right) |\delta_{t_-}| \tag{131}$$

From our previous analysis, we know that $-1 \leq \delta_{t_-} \leq 0$. In the limit of large $t$ we have

$$\lim_{t \to \infty} |\delta_t| \leq \exp\left(-16\tilde{z}_0^2/\epsilon\right) |\delta_{t_-}| \tag{132}$$

If we have the condition

$$16\tilde{z}_0^2/\epsilon \geq -\log(\epsilon) \tag{133}$$

then $\lim_{t \to \infty} |\delta_t| \leq \epsilon^2$.

If we want $\lim_{t \to \infty} y_t = -\epsilon/2 + O(\epsilon^2)$, then we need the condition

$$16\tilde{z}_0^2 \geq -\epsilon \log(\epsilon) \tag{134}$$

or equivalently $-\epsilon \log(\epsilon) < 16\tilde{z}_0^2$. Under these conditions, $\lim_{t \to \infty} \tilde{z}_t = 0$ and $\lim_{t \to \infty} y_t = -\epsilon/2 + O(\epsilon^2)$. $\quad\square$

This result can be confirmed numerically by running the dynamical equations from a variety of initializations, computing the median eigenvalue (restricted to the range $[1.9, 2.0]$), and plotting versus $\epsilon$ (Figure 10).We note that since the dynamics is slow, the ODE given by

$$\dot{\tilde{z}} = 2y\tilde{z} \tag{135}$$

$$\dot{y} = -2(4 - 3\epsilon + 4\epsilon^2)y\tilde{z}^2 - 4\epsilon\tilde{z}^2 \tag{136}$$

also obtains the same limit (Figure 10). The ODE suggests that the concentration relies on both the equal-orders in $\tilde{z}$ of the $y^0$ and $y^1$ terms, as well as a separation of timescales - $\tilde{z}$ converges to $0$ at a rate of $\epsilon$, while $y$ converges to the fixed point at a rate $\tilde{z}_t^2$. In both cases, the deviation from $-\epsilon/2$ scales as $O(\epsilon^2)$ (Figure 10, right).

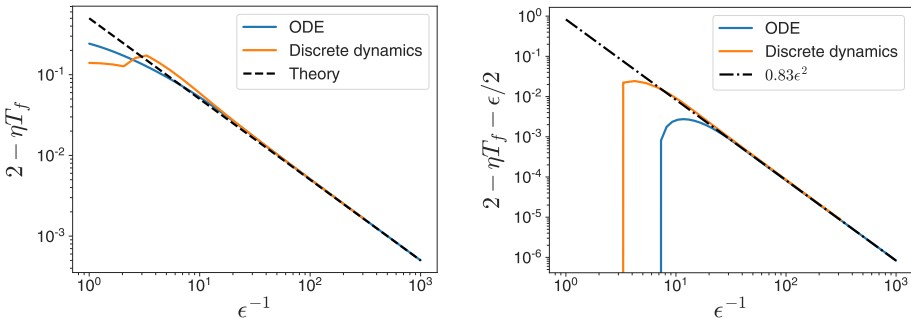

Figure 10: Final values of $y$, normalized deviation from critical value $T(0) = 2$, for discrete dynamics and ODE approximation. Deviation is well approximated by $\epsilon/2$ over a large range (left). Deviations from $\epsilon/2$ are $O(\epsilon^2)$ (right).

### B.6 PARAMETER SPACE VS. $\tilde{z} - T$ SPACE

Most of our analysis has been focused in the normalized $\tilde{z} - T$ coordinate space. In this section, we confirm that the more usual setup in parameter space is consistent with the normalized coordinate space. In particular, EOS behavior is often described by fixing an initialization, and training with different learning rates - as in Figure 1.

We can plot the dynamics of $T(0)$ for the trajectories from Figure 1. We see that for small learning rates there is convergence to $T(0) < 2$, large learning rates there's divergence, and for intermediate learning rates there is convergence to $2 - \epsilon/2$ (Figure 11).

This confirms that the theorem is useful to describe the more traditional method of discovering and exploring EOS behavior.

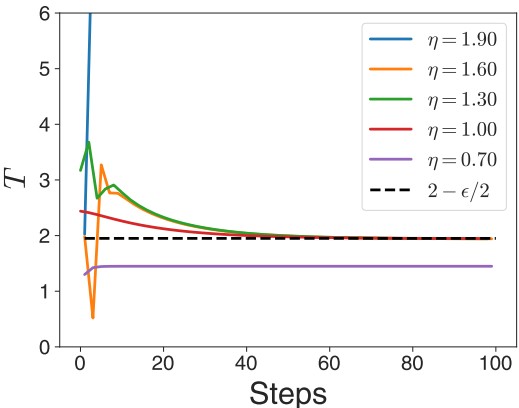

Figure 11: Dynamics of $T$ for trajectories from Figure 1. For small learning rate $\eta$, trajectories converge to $T < 2$, and for large learning rates trajectories diverge. For intermediate trajectories, we have EOS behavior, where final $T$ is predicted by Theorem 2.1.

## C QUADRATIC REGRESSION MODEL DYNAMICS

We use Einstein summation notation in this section - repeated indices on the right-hand-side of equations are considered to be summed over, unless they show up on the left-hand-side.

### C.1 PROOF OF THEOREM 3.1

Let $\mathbf{z}$, $\mathbf{J}$, and $\mathbf{Q}$ be initialized with i.i.d. random elements with $0$ mean and variance $\sigma_z^2$, $\sigma_J^2$, and $1$ respectively. Furthermore, Let the distributions be invariant to rotations in both data space and parameter space, and have finite 4th moment.

In order to understand the development of the curvature at early times, we consider coordinates which convert $\mathbf{J}$ into its singular value form. In these coordinates, we can write:

$$\mathbf{J}_{\alpha i} = \begin{cases} 0 & \text{if } \alpha \neq i \\ \sigma_\alpha & \text{if } \alpha = i \end{cases} \tag{137}$$

The singular values $\sigma_\alpha$ are the square roots of the singular values of the NTK matrix. We assume that they are ordered from largest ($\sigma_1$) to smallest in magnitude. By assumption, under this rotation the statistics of $\mathbf{z}$ and $\mathbf{Q}$ are left unchanged.

The time derivatives at $t = 0$ can be computed directly in the singular value coordinates. The first derivative is given by

$$\frac{d}{dt}\sigma_\alpha^2 = 2\sigma_\alpha\dot{\sigma}_\alpha \tag{138}$$

Using the diagonal coordinate system, we have

$$\mathrm{E}\left[\frac{d}{dt}\sigma_\alpha^2\right] = \mathrm{E}[\mathbf{Q}_{\alpha\beta j}\mathbf{J}_{\beta j}\mathbf{z}_\beta] = 0 \tag{139}$$

However, the average second derivative is positive. Calculating, we have:

$$\frac{d^2}{dt^2}\sigma_\alpha^2 = 2(\dot{\sigma}_\alpha^2 + \sigma_\alpha\ddot{\sigma}_\alpha) \tag{140}$$

We can compute the average at initialization. We have:

$$\mathrm{E}[\dot{\sigma}_\alpha^2] = \mathrm{E}[\mathbf{Q}_{\alpha\beta j}\mathbf{J}_{\beta j}\mathbf{z}_\beta\mathbf{Q}_{\alpha\delta k}\mathbf{J}_{\delta k}\mathbf{z}_\delta] = \mathrm{E}[\delta_{\beta\delta}\delta_{jk}\mathbf{J}_{\beta j}\mathbf{J}_{\delta k}\mathbf{z}_\beta\mathbf{z}_\delta] \tag{141}$$

$$\mathrm{E}[\dot{\sigma}_\alpha^2] = \mathrm{E}[\mathbf{Q}_{\alpha\beta j}\mathbf{J}_{\beta j}\mathbf{z}_\beta\mathbf{Q}_{\alpha\delta k}\mathbf{J}_{\delta k}\mathbf{z}_\delta] = \sum_j \mathrm{E}[\mathbf{J}_{\beta j}^2\mathbf{z}_\beta^2] = DP\sigma_J^2\sigma_z^2 \tag{142}$$

To compute the second term, we compute $\ddot{\mathbf{J}}_{\alpha i}$:

$$\ddot{\mathbf{J}}_{\alpha i} = -\mathbf{Q}_{\alpha ij}(\mathbf{J}_{\beta j}\dot{\mathbf{z}}_\beta + \dot{\mathbf{J}}_{\beta j}\mathbf{z}_\beta) \tag{143}$$

Expanding, we have:

$$\ddot{\mathbf{J}}_{\alpha i} = \mathbf{Q}_{\alpha ij}(\mathbf{J}_{\beta j}\mathbf{J}_{\beta k}\mathbf{J}_{\delta k}\mathbf{z}_\delta + \mathbf{Q}_{\beta jk}\mathbf{J}_{\delta k}\mathbf{z}_\delta\mathbf{z}_\beta) \tag{144}$$

In the diagonal coordinates $\mathbf{J}_{\alpha\alpha} = \sigma_\alpha$. This gives us:

$$\mathrm{E}[\sigma_\alpha\ddot{\sigma}_\alpha] = \mathrm{E}[\sigma_\alpha\mathbf{Q}_{\alpha\alpha j}\mathbf{Q}_{\beta jk}\mathbf{J}_{\delta k}\mathbf{z}_\delta\mathbf{z}_\beta] \tag{145}$$

Averaging over the $\mathbf{Q}$, we get:

$$\mathrm{E}[\sigma_\alpha\ddot{\sigma}_\alpha] = P\mathrm{E}[\sigma_\alpha\delta_{\alpha\beta}\delta_{\alpha k}\mathbf{J}_{\delta k}\mathbf{z}_\delta\mathbf{z}_\beta] = \mathrm{E}[\sigma_\alpha\mathbf{z}_\alpha\mathbf{z}_\delta\mathbf{J}_{\delta\alpha}] \tag{146}$$

Which evaluates to:

$$\mathrm{E}[\sigma_\alpha\ddot{\sigma}_\alpha] = \sigma_z^2 P\mathrm{E}[\sigma_\alpha^2] \tag{147}$$

In the limit of large $D$ and $P$, for fixed ratio $D/P$, the statistics of the Marchenko-Pastur distribution allow us to compute the derivative of the largest eigenmode as

$$\mathrm{E}[\sigma_0\ddot{\sigma}_0] = \sigma_z^2\sigma_J^2 P^2 D(1 + \sqrt{D/P})^2 \tag{148}$$

Taken together, this gives us

$$\mathrm{E}\left[\frac{d^2\lambda_{max}}{dt^2}\right] = \sigma_z^2\sigma_J^2 DP(P(1 + \sqrt{D/P})^2 + 1) \tag{149}$$

We confirm the prediction numerically in Figure 12.

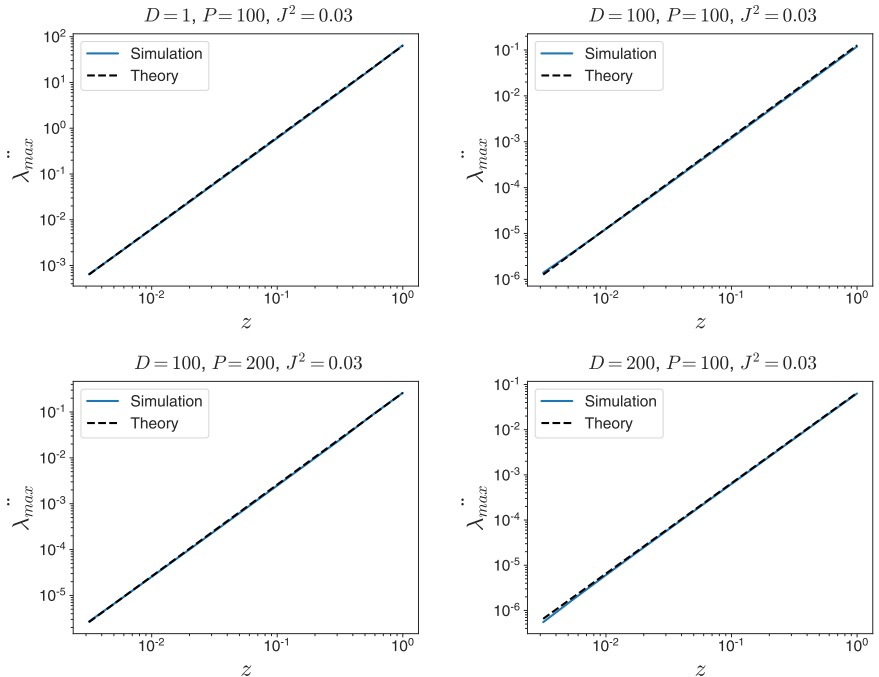

Figure 12: Average $\ddot{\lambda}_{\max}(0)$ versus $\sigma_z$, various $D$ and $P$ (100 seeds).

That is, the second derivative of the maximum curvature is positive on average. If we normalize with respect to the eigenvalue scale, in the limit of large $D$ and $P$ we have:

$$\mathrm{E}\left[\frac{d^2\lambda_{max}}{dt^2}\right]/\mathrm{E}[\lambda_{max}] = \sigma_z^2 \tag{150}$$

Therefore, increasing $\sigma_z$ increases the relative curvature of the $\lambda_{max}$ trajectory. This gives us the proof of Theorem 3.1.

This result suggests that as $\sigma_z$ increases, so does the degree of progressive sharpening. This can be confirmed by looking at GF trajectories (Figure 13). The trajectories with small $\sigma_z$ don't change their curvature much, and the loss decays exponentially at some rate. However, when $\sigma_z$ is larger, the curvature increases initially, and then stabilizes to a higher value, allowing for faster convergence to the minimum of the loss.

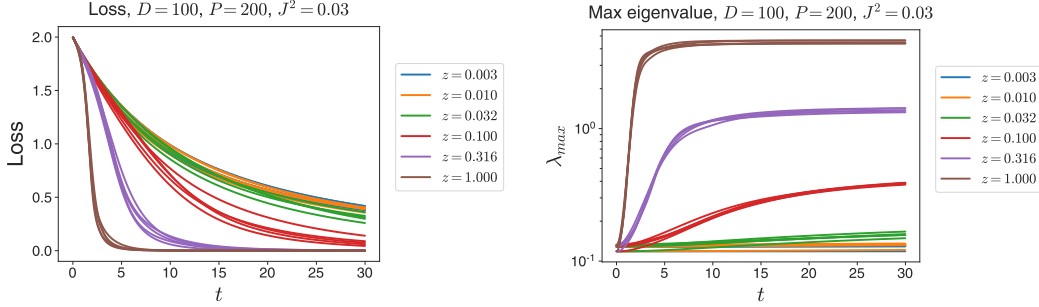

Figure 13: Gradient flow trajectories of loss and max NTK eigenvalues for quadratic regression models for varying $\sigma_z$. As $\sigma_z$ increases, $\lambda_{\max}$ changes more quickly, and is generally increasing. Models with higher $\sigma_z$ converge faster in GF dynamics.

## C.2 Timescales for gradient descent

Consider a random initialization of $\mathbf{z}$, $\mathbf{J}$, and $\mathbf{Q}$, where the terms are i.i.d. with zero mean variances $\sigma_z^2$, $\sigma_J^2$, and 1 respectively, and finite fourth moments. Furthermore, suppose that $\mathbf{z}$, $\mathbf{J}$, and $\mathbf{Q}$ are rotationally invariant in both input and output space. Under these conditions, we hope to compute

$$r_{NL}^2 \equiv \frac{\mathrm{E}[||\frac{1}{2}\eta^2\mathbf{Q}_{\alpha ij}(\mathbf{J}_{\beta i})_0(\mathbf{z}_\beta)_0(\mathbf{J}_{\delta j})_0(\mathbf{z}_\delta)_0||_2^2]}{\mathrm{E}[||\eta(\mathbf{J}_{\alpha i})_0(\mathbf{J}_{i\beta})_0(\mathbf{z}_\beta)_0||_2^2]} = \frac{1}{4}\eta^2\sigma_z^2 D^2 \tag{151}$$

at initialization, in the limit of large $D$ and $P$.

The denominator is given by:

$$\mathrm{E}[\mathbf{J}_{\alpha i}\mathbf{J}_{\beta i}(\mathbf{z}_\beta)\mathbf{J}_{\alpha j}\mathbf{J}_{\delta j}(\mathbf{z}_\delta)] = \sigma_z^2\mathrm{E}[\mathbf{J}_{\alpha i}\mathbf{J}_{\beta i}\mathbf{J}_{\alpha j}\mathbf{J}_{\delta j}\delta_{\beta\delta}] = \sigma_z^2\mathrm{E}[\mathbf{J}_{\alpha i}\mathbf{J}_{\beta i}\mathbf{J}_{\alpha j}\mathbf{J}_{\beta j}] \tag{152}$$

Evaluation gives us:

$$\mathrm{E}[\mathbf{J}_{\alpha i}\mathbf{J}_{\beta i}(\mathbf{z}_\beta)\mathbf{J}_{\alpha j}\mathbf{J}_{\delta j}(\mathbf{z}_\delta)] = \sigma_z^2(\sigma_J^4(P(P-1)D) + C_4 DP) \tag{153}$$

where $C_4$ is the 4th moment of $\mathbf{J}_{\alpha i}$. To lowest order in $D$ and $P$

$$\mathrm{E}[\mathbf{J}_{\alpha i}\mathbf{J}_{\beta i}(\mathbf{z}_\beta)\mathbf{J}_{\alpha j}\mathbf{J}_{\delta j}(\mathbf{z}_\delta)] = \sigma_z^2\sigma_J^4 DP^2 + O(DP) \tag{154}$$

Evaluating the numerator, we have:

$$\mathrm{E}[\mathbf{Q}_{\alpha ij}\mathbf{J}_{\beta i}\mathbf{z}_\beta\mathbf{J}_{\delta j}\mathbf{z}_\delta\mathbf{Q}_{\alpha mn}\mathbf{J}_{\gamma m}\mathbf{z}_\gamma\mathbf{J}_{\nu n}\mathbf{z}_\nu] = \mathrm{E}[\mathbf{J}_{\beta i}\mathbf{z}_\beta\mathbf{J}_{\delta j}\mathbf{z}_\delta\mathbf{J}_{\gamma m}\mathbf{z}_\gamma\mathbf{J}_{\nu n}\mathbf{z}_\nu](\delta_{im}\delta_{jn} + (M_4-1)\delta_{ijmn}) \tag{155}$$

where $M_4$ is the $4th$ moment of $\mathbf{Q}_{\alpha ij}$. This gives us:

$$\frac{1}{D}\mathrm{E}[\mathbf{Q}_{\alpha ij}\mathbf{J}_{\beta i}\mathbf{z}_\beta\mathbf{J}_{\delta j}\mathbf{z}_\delta\mathbf{Q}_{\alpha mn}\mathbf{J}_{\gamma m}\mathbf{z}_\gamma\mathbf{J}_{\nu n}\mathbf{z}_\nu] = \mathrm{E}[\mathbf{J}_{\beta i}\mathbf{z}_\beta\mathbf{J}_{\delta j}\mathbf{z}_\delta\mathbf{J}_{\gamma i}\mathbf{z}_\gamma\mathbf{J}_{\nu j}\mathbf{z}_\nu]+ \\ (M_4-1)\mathrm{E}[\mathbf{J}_{\beta i}\mathbf{z}_\beta\mathbf{J}_{\delta i}\mathbf{z}_\delta\mathbf{J}_{\gamma i}\mathbf{z}_\gamma\mathbf{J}_{\nu i}\mathbf{z}_\nu] \tag{156}$$

Next, we perform the $\mathbf{z}$ averages. We have

$$\frac{1}{D}\mathrm{E}[\mathbf{Q}_{\alpha ij}\mathbf{J}_{\beta i}\mathbf{z}_\beta\mathbf{J}_{\delta j}\mathbf{z}_\delta\mathbf{Q}_{\alpha mn}\mathbf{J}_{\gamma m}\mathbf{z}_\gamma\mathbf{J}_{\nu n}\mathbf{z}_\nu] = \sigma_z^4\mathrm{E}[\mathbf{J}_{\beta i}\mathbf{J}_{\delta j}\mathbf{J}_{\gamma i}\mathbf{J}_{\nu j}](\delta_{\beta\delta}\delta_{\gamma\nu} + \delta_{\beta\gamma}\delta_{\delta\nu} + \delta_{\beta\nu}\delta_{\delta\gamma}) \\ + (C_4-\sigma^4)\mathrm{E}[\mathbf{J}_{\beta i}\mathbf{J}_{\delta j}\mathbf{J}_{\gamma i}\mathbf{J}_{\nu j}]\delta_{\beta\delta\gamma\nu} \\ + (M_4-1)\sigma_z^4\mathrm{E}[\mathbf{J}_{\beta i}\mathbf{J}_{\delta i}\mathbf{J}_{\gamma i}\mathbf{J}_{\nu i}](\delta_{\beta\delta}\delta_{\gamma\nu} + \delta_{\beta\gamma}\delta_{\delta\nu} + \delta_{\beta\nu}\delta_{\delta\gamma}) \\ + (M_4-1)(C_4-\sigma^4)\mathrm{E}[\mathbf{J}_{\beta i}\mathbf{J}_{\delta i}\mathbf{J}_{\gamma i}\mathbf{J}_{\nu i}]\delta_{\beta\delta\gamma\nu} \tag{157}$$

where $C_4$ is the 4th moment of $\mathbf{z}$. Simplification gives us:

$$\frac{1}{D}\mathrm{E}[\mathbf{Q}_{\alpha ij}\mathbf{J}_{\beta i}\mathbf{z}_\beta\mathbf{J}_{\delta j}\mathbf{z}_\delta\mathbf{Q}_{\alpha mn}\mathbf{J}_{\gamma m}\mathbf{z}_\gamma\mathbf{J}_{\nu n}\mathbf{z}_\nu] = \sigma_z^4(\mathrm{E}[\mathbf{J}_{\beta i}\mathbf{J}_{\beta j}\mathbf{J}_{\delta i}\mathbf{J}_{\delta j}] + \mathrm{E}[\mathbf{J}_{\beta i}\mathbf{J}_{\delta j}\mathbf{J}_{\beta i}\mathbf{J}_{\delta j}] + \mathrm{E}[\mathbf{J}_{\beta i}\mathbf{J}_{\delta j}\mathbf{J}_{\delta i}\mathbf{J}_{\beta j}]) \\ + (C_4-\sigma_z^4)\mathrm{E}[\mathbf{J}_{\beta i}\mathbf{J}_{\beta j}\mathbf{J}_{\beta i}\mathbf{J}_{\beta j}] \\ + (M_4-1)\sigma_z^4(\mathrm{E}[\mathbf{J}_{\beta i}\mathbf{J}_{\beta i}\mathbf{J}_{\gamma i}\mathbf{J}_{\gamma i}] + \mathrm{E}[\mathbf{J}_{\beta i}\mathbf{J}_{\delta i}\mathbf{J}_{\beta i}\mathbf{J}_{\delta i}] + \mathrm{E}[\mathbf{J}_{\beta i}\mathbf{J}_{\delta i}\mathbf{J}_{\delta i}\mathbf{J}_{\beta i}]) \\ + (M_4-1)(C_4-\sigma^4)\mathrm{E}[\mathbf{J}_{\beta i}\mathbf{J}_{\beta i}\mathbf{J}_{\beta i}\mathbf{J}_{\beta i}] \tag{158}$$

For large $D$ and $P$, the final three terms are asymptotically smaller than the first term. Evaluating the first term, to leading order we have:

$$\frac{1}{D}\mathrm{E}[\mathbf{Q}_{\alpha ij}\mathbf{J}_{\beta i}\mathbf{z}_\beta\mathbf{J}_{\delta j}\mathbf{z}_\delta\mathbf{Q}_{\alpha mn}\mathbf{J}_{\gamma m}\mathbf{z}_\gamma\mathbf{J}_{\nu n}\mathbf{z}_\nu] = \sigma_z^4\sigma_J^4(2DP^2 + 2D^2P + D^2P^2) + O(D^2P + DP^2) \tag{159}$$

$$\mathrm{E}[\mathbf{Q}_{\alpha ij}\mathbf{J}_{\beta i}\mathbf{z}_\beta\mathbf{J}_{\delta j}\mathbf{z}_\delta\mathbf{Q}_{\alpha mn}\mathbf{J}_{\gamma m}\mathbf{z}_\gamma\mathbf{J}_{\nu n}\mathbf{z}_\nu] = \sigma_z^4\sigma_J^4 D^3P^2 + O(D^3P + D^2P^2) \tag{160}$$

This gives us:

$$r_{NL}^2 = \frac{1}{4}\frac{\sigma_z^4\sigma_J^4 D^3P^2}{\sigma_z^2\sigma_J^4 DP^2} = \frac{1}{4}\sigma_z^2 D^2 \tag{161}$$

to leading order, in the limit of large $D$ and $P$.

## C.3 Dependence on $D$ and $P$

We can see empirically that the sharpening is more pronounced in the overparameterized regime where $D > P$. Using the trajectories from Figure 4, we can make a scatter plot of the normalized maximum NTK eigenvalues $\eta\lambda_{max}$ at both initialization and the final point of the dynamics (Figure C.3). In all cases, a variety of initializations ($x$-axis) lead to final values which concentrate around 2 ($y$-axis).

We can see that the concentration is tightest for the overparameterized regime where $P > D$ (right plot). We hypothesize that for large $D$ and $P$, the EOS behavior is stronger and more likely to happen when $P > D$. We leave futher exploration of this hypothesis for future work.

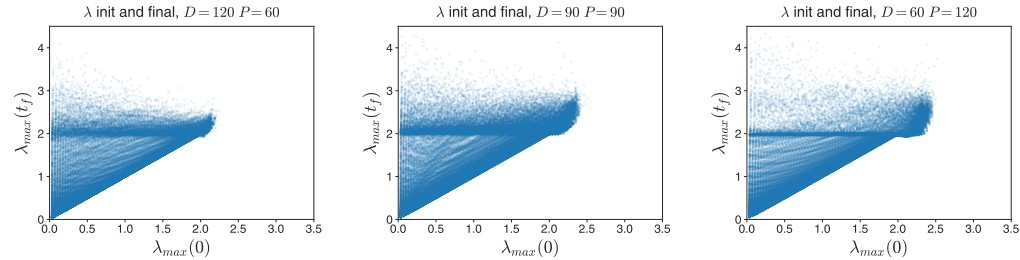

Figure 14: Scatter plots of initial vs. final normalized maximum eigenvalues $\eta\lambda_{max}$ for quadratic regression models. Trajectories are taken from the data used to generate Figure 4. For large $D$ and $P$, as the model becomes overparameterized ($P > D$), a subset of trajectories show tighter EOS behavior where $\eta\lambda_{max}$ concentrates close to 2 for a variety of initializations.

# D Analysis of real models

## D.1 Dynamics of $y$ in CIFAR10 model

The dynamics of $y$ in the CIFAR10 model analyzed in Section 4 are more complicated than the $z_1$ dynamics. We see from Figure 5 that there is a $z_1$ and $y$-independent component of the two-step change in $y$. We can approximate this change $b$ by computing the average value of $y_{t+2} - y_t$ for small $z_1$ (taking $z_1 < 10^{-4}$ in this case). We can then subtract off $b$ from $y_{t+2} - y_t$, and plot the remainder against $z_t^2$ (Figure 15 left). We see that $y_{t+2} - y_t - b$ is negatively correlated with $z_t^2$, particularly for large $z_t$. However, $y_{t+2} - y_t$ is clearly not simply function of $z_1$.

The two-step model dynamics could be written as $(ay + c)\tilde{z}^2$. If we plot $(y_{t+2} - y_t - b)/z_1^2$ versus $y_t$, we again don't have a single-valued function (Figure 15, right). Therefore, the functional form of $y_{t+2} - y_t$ is not given by $b + ayz_1^2 + cz_1^2$.

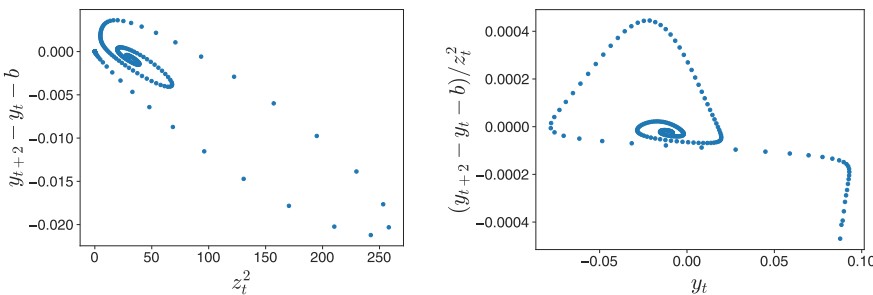

Figure 15: Gradient flow trajectories of loss and max NTK eigenvalues for quadratic regression models for varying $\sigma_z$. As $\sigma_z$ increases, $\lambda_{max}$ changes more quickly, and is generally increasing. Models with higher $\sigma_z$ converge faster in GF dynamics.

### D.2 QUADRATIC EXPANSION OF 2-CLASS CIFAR MODEL

We trained a CIFAR model using the first two classes only with $5000$ datapoints using the Neural Tangents library (Novak et al., 2019) - which let us perform 2nd and 3rd order Taylor expansions of the model at arbitrary parameters. The models were 2-hidden layer fully-connected networks, with hidden width $256$ and Erf non-linearities. Models were initialized with the NTK parameterization, with weight variance $1$ and bias variance $0$. The targets were scalar valued - $+1$ for the first class, $-1$ for the second class. A learning rate of $0.003204$ was used in all experiments. All plots were made using float-64 precision.

Taking a quadratic expansion at initialization, we see that the loss tracks the full model for the first $1000$ steps in this setting (Figure 16, left), but misses the edge-of-stability behavior. We use Neural Tangents to efficiently compute the NTK to get the top eigenvalue $\lambda_1$ (and consequently, $y$). We can also compute $z_1$ by computing the associated eigenvector $\mathbf{v}_1$ and projecting residuals $\mathbf{z}$. If the quadratic expansion is taken closer to the edge of stability, the dynamics of $z_1$ well approximates the true $z_1$ dynamics, up to a shift associated with exponential growth of $z_1$ occurring at different times (Figure 16, middle). We see that the shape of the first peak in $|z_1|$ is the same for the full model and the quadratic model, but the subsequent oscillations are faster and more quickly damped in the full model. This suggests that the initial EOS behavior may be captured by the quadratic model, but the detailed dynamics require an understanding of higher order terms. For example, the 3rd order Taylor expansion improves the prediction of the magnitude and period of the oscillations, but still misses key quantitative features (Figure 16, right).

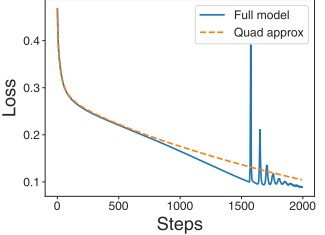 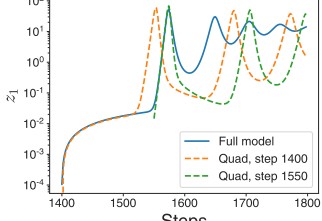 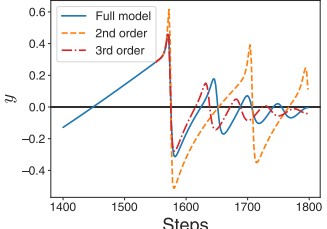

Figure 16: Quadratic expansion of FCN model trained on two-class CIFAR. Expanding at initialization gives good approximation to full model for $1000$ steps, after which EOS behavior occurs in full model but not approximate one (left). When $z_1$ is small, quadratic model tracks full model; however, initial exponential increase may happen earlier in approximate model (middle). Magnitude of $z_1$ has larger oscillations in full model compared to approximate model. Third-order Taylor expansion better captures magnitude and period of oscillations, but still misses quantitative features (right).

