# OpenReview forum: "A second order regression model shows edge of stability behavior"
_ICLR.cc/2023/Conference — Submitted to ICLR 2023_

### Official Review · Reviewer_n7HJ · 2022-10-25

**Confidence:** 4
**Correctness:** 3
**Technical Novelty And Significance:** 2
**Empirical Novelty And Significance:** 1
**Recommendation:** 6

**Clarity, Quality, Novelty And Reproducibility:**

> gradient descent (GD) trajectories where the maximum eigenvalue of the NTK remains close to the critical value 2/η

> we define edge of stability with respect to the maximum NTK eigenvalue

> this is equivalent to the maximum eigenvalue of the loss Hessian used in Cohen et al. (2022a) as the model converges to a stationary point (Jacot et al., 2020).

This argument appears novel to me.
I searched in Cohen et al. (2022a) and (Jacot et al., 2020), but didn't find any statement on equivalence between Hessian and NTK.

Next I will show the difference between them.

Consider loss $L(\theta)=\sum_{i=1}^n l_i(f_i(\theta))$, where $f_i:\mathbb{R}^d\to\mathbb{R},l_i:\mathbb{R}\to\mathbb{R}$ the Hessian is $H=\sum_{i=1}^n(\nabla f_i(\theta) l_i'' \nabla f_i(\theta)^\top + l_i'\nabla^2 f_i(\theta) )$, where $\nabla f_i(\theta)$ is a column vector and $\nabla^2 f_i(\theta)$ is a $d$ by $d$ matrix.

On the other hand, NTK is defined as a $n$ by $n$ matrix $\Theta_{i,j}=\nabla f_i(\theta)^\top \nabla f_j(\theta)$.

Clearly, Hessian and NTK are different.

Even if authors wish to establish equivalence to spectrum only, heavy assumptions are needed. First, we need to assume $l_i''=1$ for all $i$, which means we can only use square loss. Second, we need to to make sure $\sum_i l_i' \nabla^2 f_i(\theta)=0$. This is not necessarily true at stationary point, because first order stationary condition only guarantees $\sum_i l_i' \nabla f_i(\theta)=0$.

Besides, I don't think stationary condition should be used at all while studying EOS, because EOS only happens before we reach stationary.

> Figure 1 ... The GD trajectories converge to minima with larger curvature than at initialization ...

The figure doesn't show curvature directly.
It only shows loss in log scale, and it hard for me to see which part has larger curvature.
Thus, it is not clear to me whether GD trajectories really converge to minima with larger curvature.


> Figure 1.

What are the initial points of these GD trajectories?

> we can see that $\operatorname{sign}(\Delta T(0))=\operatorname{sign}\left(T_t(0)-4\right)$, as in Lewkowycz et al. (2020)

Could authors point out which part in Lewkowycz et al. (2020) is relevant?

> so convergence requires strictly decreasing curvature

The reason is not totally clear to me. Could authors provide further explanation?

> Figure 2

Figure 2 middle shows that curvature should first increase, and then decrease and converge to a value that is slightly smaller than critical curvature.

However, Figure 1 left shows first oscillation and then monotone increasing curve.

Why there is such mismatch between theory and experiment?

> which reveals that the spectrum does not shift much ... This suggests that Q doesn’t change much as these EOS dynamics are displayed

In order to constitute a sufficient condition, I believe authors also need to show the eigenvectors doesn't change much.
Is there any intuition why eigenvectors don't need to be verified?

> (Section 2) $y_t =T_t(0)-f_{\tilde{z}}\left(\tilde{z}_t\right)$ (Section 4) $y=\lambda_1\eta-2$... we see that the two-step dynamics of $z_1$ is well approximated by $2yz$ (Figure 6, middle). This is the same form that the dynamics of $\tilde{z}$ takes in our simplified model.

The definitions of $y$ in theory and in experiment are very different. Is there any correspondence between two definitions?

**Strength And Weaknesses:**

Strength:

The topic should be interesting to optimization community.

Weakness:

This article tampered with the definition of EOS by changing Hessian into NTK. More specific discussion is in following section.

Many claims is not supported. More specific discussion is in following section.

There is mismatch between experiment and theory even in simple quadratic model.

**Summary Of The Paper:**

This paper studies the dynamics of a quadratic model trained with quadratic loss.
The effective dynamics of NTK is derived.
Authors argue that edge of stability behavior in neural networks is correlated with the behavior in quadratic regression models.

**Summary Of The Review:**

This paper focuses on quadratic model with quadratic loss and derive effective dynamics of loss and NTK for the model.
The topic is interesting.
However, there are many unjustified claims. Moreover, the experiments and theory doesn't match well.

---

> ### Author Response · Authors · 2022-11-09
> **Response to Reviewer n7HJ - NTK vs. Hessian spectrum**
>
> We thank the reviewer for their detailed feedback. We will respond to the comments individually, in order to make 3 points:
> * Studying edge of stability using NTK spectrum is valid in our setting.
> * Our main claims are well supported by our theory and experiments.
> * Experiment and theory matches well for the simplified quadratic model.
>
> On the usage of the NTK spectrum versus the Hessian spectrum:
>
> Thanks to the reviewer for this subtle point: we agree that our presentation/motivation for the differences is lacking in our original draft.
>
> Our main point is that, for squared loss, both theory and experiments suggest that EOS effects happen with respect to the NTK spectrum as well as for the Hessian spectrum. We chose to focus on the NTK version of the effect as it allowed us to understand the behavior theoretically.
>
> The EOS effects were well-categorized as an empirical phenomenon in Cohen et. al. 2022. While studying both the quadratic model and the CIFAR model we present in the paper, we discovered that the largest NTK eigenvalue displays EOS behavior as well - where the edge is computed with respect to Equation 25 when Q = 0. Through private communication with the authors of Cohen et. al., it was acknowledged that EOS with respect to the max NTK eigenvalue is an interesting and suitable alternative for study.
>
> For squared loss, the spectrum of the NTK is the same as the Gauss-Newton approximation to the Hessian. This approximation becomes exact in the limit as the gradient goes to zero. This means that for the one datapoint, two-parameter model, Theorem 2.1 (proven explicitly for the NTK spectrum) holds for the Hessian as well - since in that theorem, the EOS behavior is present as z converges to the minimum. The theorem can also be proven by computing the Hessian eigenvalues directly, which results in messier algebra.
>
> _Even if authors wish to establish equivalence to spectrum only, heavy assumptions are needed... Besides, I don't think stationary condition should be used at all while studying EOS, because EOS only happens before we reach stationary._
>
> As the reviewer points out, EOS behavior often occurs far from convergence - meaning the exact correspondence is not valid generally. However, as we show in Figure 5 for the CIFAR model (taken from Cohen et. al.), EOS behavior happens in the NTK spectrum as well. Therefore even if the two spectra are not equivalent, EOS behavior in the NTK exists and is interesting to study.
>
> Studying the NTK spectrum is best with squared loss (as the reviewer correctly points out). However, following Cohen et. al. we focused on the case of squared loss because the phenomenology of sharpening and EOS is more complicated in the non-squared loss setting. We leave proving rigorous theorems about sharpening and EOS in more complicated settings for future work.
>
> We have added elements of this discussion to Appendix A.1 in the new draft; we welcome any further feedback on how to best communicate the relationship between the two approaches.

---

> > ### Author Response · Authors · 2022-11-09
> > **Response (continued)**
> >
> > We address the rest of the comments below.
> >
> > _[Figure 1] doesn't show curvature directly. It only shows loss in log scale, and it hard for me to see which part has larger curvature. Thus, it is not clear to me whether GD trajectories really converge to minima with larger curvature._
> >
> > The minima form an X-shape defined by $|\theta_2| = \epsilon^{-1/2}|\theta_1|$. Away from the origin, the width of the dark blue region around the minima gives an indication of the curvature - smaller width corresponds to higher curvature. Since the loss surface is 4-th order in the parameters, the curvature at the origin (0, 0) is in fact 0.
> >
> >
> > _Figure 2 middle shows that curvature should first increase, and then decrease and converge to a value that is slightly smaller than critical curvature._
> >
> > _However, Figure 1 left shows first oscillation and then monotone increasing curve._
> >
> > _Why there is such mismatch between theory and experiment?_
> >
> > We apologize for the confusion between Figures 1 and 2. We note that they show the same dynamical equations, plotted in different coordinate systems, with different initializations/learning rates - so there is no mismatch between the two dynamics. We have added Appendix B.6 to elaborate on the relationship between the two.
> >
> > If we focus on the orange curve in Figure 1, left, in the first couple of steps the trajectory goes from the right side of the X-shape of the minima to the left side - in a very discontinuous jump. This corresponds to an initial increase in the curvature. There is then an oscillatory approach to the fixed point near the edge of stability. In Figure 1, right, we plot every-other iterate of the dynamics, which shows a relatively smooth descent into the minima at the edge of stability after the transient increase in curvature. This corresponds to the dynamics shown in Figure 2 middle, which is also plotted every other iterate - again getting rid of the oscillations.
> >
> > We can see the correspondence directly by computing $T$ for the trajectories in Figure 1 and plotting every other iterate - as seen in Figure 10 of the latest draft. For small learning rates, there is immediate convergence, with $T<2$; for large learning rates, the dynamics diverges. For intermediate learning rates, we see exactly the EOS behavior predicted by Theorem 2.1, with good agreement to the predicted converged value $2-\epsilon/2$. This is because, as mentioned previously, the dynamical equations are exactly the same in the two figures.
> >
> > _Could authors point out which part in Lewkowycz et al. (2020) is relevant to:_
> >
> > "so convergence requires strictly decreasing curvature"
> >
> > _The reason is not totally clear to me. Could authors provide further explanation?_
> >
> > When describing the catapult phase, Lewkowycz et al. say:
> >
> > "First, note that $\eta \lambda_{0} − 4 < 0$ by assumption, and therefore the (additive) kernel updates are negative for all $t$"
> >
> > One can also look directly at Equation 7 and see that the sign of $\Delta \lambda$ only depends on $\lambda$. When $\eta\lambda$ (equivalent to our $T$) is greater than $4$, $\lambda$ is strictly increasing; for $\eta\lambda<4$, $\lambda$ is strictly decreasing. Therefore $\lambda$ (and $T$) is either strictly increasing, constant, or strictly decreasing.
> >
> > Regarding the behavior near convergence: Equation 6 converges to $f = 0$ only if $\eta\lambda < 2$ at late times. If $\lambda$ is strictly increasing, $\eta\lambda>2$ for all times, so there is no convergence in Equation 6. If $\Delta \lambda = 0$, $\eta\lambda>2$ as well so there is no convergence. Therefore, convergence requires strictly decreasing $\lambda$.
> >
> > _In order to constitute a sufficient condition, I believe authors also need to show the eigenvectors doesn't change much. Is there any intuition why eigenvectors don't need to be verified?_
> >
> > We agree that a detailed analysis/proof would show bounds on the eigenvector changes, as well as a more detailed analysis of the empirical $Q$. However we included the results on the change in $Q$ in order to give some grounded hypotheses related to the phenomenology, not to provide a definitive proof of the results.
> >
> > _The definitions of y in theory and in experiment are very different. Is there any correspondence between two definitions?_
> >
> > When $z$ goes to $0$ in the theoretical model, the definition of $y$ matches the definition in the experiments. In the experiments, we chose the simpler definition of $y = \eta\lambda_{max}-2$, as we do not yet have a theory to predict the $z$-dependence which would make the dynamics simplify, as we do in the low-dimensional case.
> >
> > We hope this clears up any concerns about the validity of our approach and results; let us know if anything else can be cleared up.

---

> > > ### Comment · Reviewer_n7HJ · 2022-11-11
> > > **Thanks for the response**
> > >
> > > I thank authors for their reply and revision, which resolve many of my concerns.
> > >
> > > > Figure 2 middle shows that curvature should first increase, and then decrease and converge to a value that is slightly smaller than critical curvature.
> > > However, Figure $\textcolor{red}{2}$ left shows first oscillation and then monotone increasing curve.
> > >
> > > Sorry, I made a typo in the original comment. I still cannot understand why there is a difference between Figure 2 middle and Figure 2 left.

---

> > > > ### Author Response · Authors · 2022-11-11
> > > > **Updated Figure 2 left**
> > > >
> > > > Thanks for the catch! Upon further investigation, we plotted every other iterate for the negative values of z - which in our theorem approach the limit from below - rather than the positive ones, which approach from above (and what we used in the middle and right plot). We have uploaded a new version of the draft that fixes this issue (by plotting the even-numbered steps instead of the odd-numbered steps). We hope this resolves the concern.

---

> > > > > ### Comment · Reviewer_n7HJ · 2022-11-19
> > > > > **Experiment is sensitive to initialization**
> > > > >
> > > > > Thanks for the revision!
> > > > > I think the using odd-even swap to explain the the increasing curvature in original Figure 2 left makes sense to me.
> > > > > However, I would like to be more cautious with experimental changes. So I tried to reproduce the experiment myself.
> > > > > The first issue I spot is that results like Figure 2 left can only be obtained by carefully selected initialization.
> > > > >
> > > > > | initial point | curvature limit |
> > > > > | ----------- | ----------- |
> > > > > | (1.5, −6) | diverge |
> > > > > | (1.5, −5) | 1.9401252 |
> > > > > | (2, −4.32) | 1.3691394 |
> > > > > | **(1.5, −4.32)** | 1.9456072 |
> > > > > | (1.5, −4) | 1.7666852 |
> > > > > | (1.5, −3) | 0.9694365 |
> > > > >
> > > > > Here I used step size = 1.0, therefore the critical curvature is 2.0, and corresponds to purple curve in Figure 2 left.
> > > > >
> > > > > This paper used initialization (1.5, −4.32), and shows that curvature converge to a value close to 2.0.
> > > > > I tried several other values, and many initialization lead to convergence to a value far away from 2.0.
> > > > >
> > > > > The above initialization sensitivity analysis shows that in second order regression model, only a small portion of the initialization demonstrates EOS behavior.

---

> > > > > > ### Comment · Reviewer_n7HJ · 2022-11-19
> > > > > > **Related Theory Weakness**
> > > > > >
> > > > > > The theory in this paper doesn't provide any description on what initialization is required.
> > > > > > The Theorem 2.1 reads like $$\exists \text{initialization}, \exists \text{step size:} \eta, \operatorname{s.t.} \lim_{t\to\infty}\lambda_{max} \in [2/\eta-\epsilon,2/\eta].$$
> > > > > >
> > > > > > This is not an informative result if it only applies to "certain initialization". For example, $$\exists \text{initialization}, \exists \text{step size:} \eta, \operatorname{s.t.} \lim_{t\to\infty}\lambda_{max} \notin [2/\eta-\epsilon,2/\eta]$$
> > > > > > is also true (by considering a neighborhood around origin).
> > > > > >
> > > > > > I believe authors analysis in this paper could lead to stronger result, which include a premise on initialization. Theorem B.3 seems to try to solve this problem. However, the proof of Theorem B.3 need to be amended as it uses an undefined relationship "≪", so the proof is not rigorous.

---

> > > > > > > ### Author Response · Authors · 2022-11-19
> > > > > > > **Response to theory weakness**
> > > > > > >
> > > > > > > Thanks for the catch! We agree that as stated, Theorem 2.1 would be quite trivial to prove!
> > > > > > >
> > > > > > > Our result actually is stronger; in our attempt simplify our theorem statement we dropped one of the key messages - the bound corresponds to a _uniform_ region in function space - independent of $\epsilon$.
> > > > > > >
> > > > > > > As shown in the proofs in Appendix B, the key point is that in $\tilde{z}-T$ space (or more accurately, $\tilde{z}-y$ space), there is a provably EOS neighborhood which is UNIFORM in $\epsilon$. This means that as $\epsilon$ is adjusted (decreased, for example), the region might change in parameter x learning rate space ($\theta-\eta$ space); however, the picture in the normalized model output ($\tilde{z}$)-normalized curvature ($T$) space does not change with $\epsilon$!
> > > > > > >
> > > > > > > This means that the theorem is non-trivial; we can fix a point in $\tilde{z}-T$ space, decrease $\epsilon$, and the theorem still holds. This means that there are initializations with $T-2$ much larger than $\epsilon$ which nonetheless converge to $O(\epsilon)$ of $T = 2$ - a non-trivial result. Similarly, due to the relationship between $\eta$ and the conversion to $\tilde{z}-T$ space, the range of learning rates over which the theorem holds remains much larger than $\epsilon$ as $\epsilon$ goes to $0$.
> > > > > > >
> > > > > > > We have update the theorem to add the additional point of uniformity in $\tilde{z}-y$ space. We are happy to add further discussion of the shape of the valid regions and the relationship to $\epsilon$ if that would further improve the presentation.

---

> > > > > > > > ### Comment · Reviewer_n7HJ · 2022-11-19
> > > > > > > > **Thanks for the revision**
> > > > > > > >
> > > > > > > > The authors' earnest response addressed all my concerns. Therefore I change my recommendation accordingly.
> > > > > > > >
> > > > > > > > I still recommend authors to amend Theorem B.3 even though it is not the main result of this paper.

---

> > > > > > ### Comment · Reviewer_n7HJ · 2022-11-19
> > > > > > **Code I Used**
> > > > > >
> > > > > > ```
> > > > > > import jax
> > > > > > import jax.numpy as np
> > > > > >
> > > > > > P = 2
> > > > > > Q = np.array([[1, 0], [0, 0]]) + np.array([[0, 0], [0, -0.1]])
> > > > > > E = 0
> > > > > >
> > > > > >
> > > > > > def f(theta):
> > > > > >     z = (np.dot(theta, Q @ theta) - E)/2
> > > > > >     return z**2 / 2
> > > > > >
> > > > > > def one_gd(f, p0, step):
> > > > > >     return p0 - step * jax.grad(f)(p0)
> > > > > >
> > > > > >
> > > > > > def gd(p, step):
> > > > > >     next_p = p - step * jax.grad(f)(p)
> > > > > >     fpp = f(next_p)
> > > > > >     lam, _ = np.linalg.eigh(jax.hessian(f)(next_p))
> > > > > >     return next_p, (next_p, fpp, lam)
> > > > > >
> > > > > > print(f"critical value: {2/step}")
> > > > > >
> > > > > >
> > > > > > def critical(p0):
> > > > > >     p0 = np.array(p0)
> > > > > >     step = 1.0
> > > > > >     _, (ps, fs, lams) = jax.lax.scan(gd, p0, np.repeat(step, 500))
> > > > > >     print(p0, float(lams[-1, 1]))
> > > > > >
> > > > > >
> > > > > > critical([1.5, -6])
> > > > > > critical([1.5, -5])
> > > > > > critical([2.0, -4.32])
> > > > > > critical([1.5, -4.32])
> > > > > > critical([1.5, -4])
> > > > > > critical([1.5, -3])
> > > > > > ```
> > > > > >
> > > > > > Output:
> > > > > > ```
> > > > > > critical value: 2.0
> > > > > > [ 1.5 -6. ] nan
> > > > > > [ 1.5 -5. ] 1.9401252269744873
> > > > > > [ 2.   -4.32] 1.3691394329071045
> > > > > > [ 1.5  -4.32] 1.9456071853637695
> > > > > > [ 1.5 -4. ] 1.7666852474212646
> > > > > > [ 1.5 -3. ] 0.969436526298523
> > > > > > ```

---

> > > > > > > ### Author Response · Authors · 2022-11-19
> > > > > > > **Response to experiments**
> > > > > > >
> > > > > > > Thanks for going the extra mile to do the experiments yourself! The code looks correct to us.
> > > > > > >
> > > > > > > The short answer is: the key is to focus on (initialization, learning rate) _pairs_. For fixed learning rate, an individual initialization in this model may not be at the edge of stability; however, there is often a learning rate which will lead to EOS behavior. We note that this point can be seen in "real" ML models as well, and is discussed in Cohen 2022!
> > > > > > >
> > > > > > > In fact, the easiest way to find EOS behavior in these models is to 1. pick an initialization and 2. sweep through learning rates. This is the experimental approach followed in most EOS studies. Below this comment we have added a small modification to your code which demonstrates this fact with the initializations you've chosen.
> > > > > > >
> > > > > > > The key is that if the learning rate is small enough, the model converges without having to cross the stability threshold (lam_max < 2/step_size at all times). This leads, naturally, to no EOS behavior. To get EOS behavior, the learning rate needs to be high enough to trigger the dynamics to cross the stability threshold, but small enough that the dynamics doesn't diverge. The main point of our theorem is to show that such a region exists, where we can prove that the final lam_max is close to 2/step_size.

---

> > > > > > > ### Author Response · Authors · 2022-11-19
> > > > > > > **Learning rate sweep code**
> > > > > > >
> > > > > > > Replace the `critical` function with:
> > > > > > >
> > > > > > > ```
> > > > > > > def critical(p0, step):
> > > > > > >     p0 = np.array(p0)
> > > > > > >     _, (ps, fs, lams) = jax.lax.scan(gd, p0, np.repeat(step, 500))
> > > > > > >     return float(lams[-1, 1])
> > > > > > > ```
> > > > > > >
> > > > > > > Recreating your results:
> > > > > > >
> > > > > > > ```
> > > > > > > step = 1.0
> > > > > > >
> > > > > > > p0s = [[1.5, -6],
> > > > > > >        [1.5, -5],
> > > > > > >        [2.0, -4.32],
> > > > > > >        [1.5, -4.32],
> > > > > > >        [1.5, -4],
> > > > > > >        [1.5, -3],]
> > > > > > >
> > > > > > > print(f"critical value: {2/step}")
> > > > > > >
> > > > > > > for p0 in p0s:
> > > > > > >   lam_final = critical(p0, step)
> > > > > > >   print(f"p0: {p0}, lam_max: {lam_final:.2f}")
> > > > > > > ```
> > > > > > >
> > > > > > > Doing a learning rate sweep instead:
> > > > > > >
> > > > > > > ```
> > > > > > > steps = np.linspace(0.5, 3.0, 10)
> > > > > > >
> > > > > > >
> > > > > > > p0 = p0s[0]
> > > > > > >
> > > > > > > print(f"p0: {p0}")
> > > > > > >
> > > > > > > for step in steps:
> > > > > > >   lam_final = critical(p0, step)
> > > > > > >   print(f"step: {step:.2f} step*lam: {step*lam_final:.2f}")
> > > > > > >
> > > > > > > p0 = p0s[-1]
> > > > > > >
> > > > > > > print(f"p0: {p0}")
> > > > > > >
> > > > > > > for step in steps:
> > > > > > >   lam_final = critical(p0, step)
> > > > > > >   print(f"step: {step:.2f} step*lam: {step*lam_final:.2f}")
> > > > > > > ```
> > > > > > >
> > > > > > > which gives output:
> > > > > > >
> > > > > > > ```
> > > > > > > p0: [1.5, -6]
> > > > > > > step: 0.50 step*lam: 1.86
> > > > > > > step: 0.78 step*lam: 1.94
> > > > > > > step: 1.06 step*lam: nan
> > > > > > > step: 1.33 step*lam: nan
> > > > > > > step: 1.61 step*lam: nan
> > > > > > > step: 1.89 step*lam: nan
> > > > > > > step: 2.17 step*lam: nan
> > > > > > > step: 2.44 step*lam: nan
> > > > > > > step: 2.72 step*lam: nan
> > > > > > > step: 3.00 step*lam: nan
> > > > > > > p0: [1.5, -3]
> > > > > > > step: 0.50 step*lam: 0.53
> > > > > > > step: 0.78 step*lam: 0.80
> > > > > > > step: 1.06 step*lam: 1.00
> > > > > > > step: 1.33 step*lam: 1.02
> > > > > > > step: 1.61 step*lam: 1.20
> > > > > > > step: 1.89 step*lam: 1.86
> > > > > > > step: 2.17 step*lam: 1.94
> > > > > > > step: 2.44 step*lam: 1.94
> > > > > > > step: 2.72 step*lam: nan
> > > > > > > step: 3.00 step*lam: nan
> > > > > > > ```

---

> > > > > > > > ### Author Response · Authors · 2022-11-19
> > > > > > > > **Learning rate sweep code (cont.)**
> > > > > > > >
> > > > > > > > In the above code we print out $\eta\lambda_{max}$, since this quantity is near the universal value of $2$ for EOS behavior.
> > > > > > > >
> > > > > > > > As you can see, by taking a diverged initialization and decreasing the learning rate, we first converge at edge of stability; taking a case where the initial setup gave convergence below the edge, increasing the learning rate eventually brings it to the edge of stability before divergence.
> > > > > > > >
> > > > > > > > If it would be helpful to add clarification on this point (fixed point, sweep learning rate vs. fixed learning rate, select points), and its relationship to the normalized z-T variables, we would be happy to modify the main text/appendix as needed. This is a subtle point and we thank the reviewer for their engagement on the issue.

---

> > > > ### Author Response · Authors · 2022-11-15
> > > > **Let us know of any other concerns**
> > > >
> > > > Please let us know if there are any other concerns we can clear up. If we have resolved your concerns, we would appreciate it if you would consider revisiting your score. Thanks!

---

> > > > > ### Comment · Reviewer_n7HJ · 2022-11-19
> > > > > **Another question about oscillation in original Figure 2 left**
> > > > >
> > > > > Sorry for the late question.
> > > > >
> > > > > In reproducing Figure 2 on the left, I can observe that the curvature increases in odd/even order. However, in the original (before revision) of Figure 2 on the left, there are strong oscillations in the first 30 iterations, which I could not observe in the replication.
> > > > >
> > > > > Is there any explanation for this oscillation?

---

> > > > > > ### Author Response · Authors · 2022-11-19
> > > > > > **Oscillation explanation**
> > > > > >
> > > > > > The additional oscillations were due to an issue with the initialization in the original plot. Due to an error in our code, the initialization originally plotted corresponded to a value of $T$ of 3.5 - a very large initialization which leads to very large jumps at early times, even when plotting every other step. This is also far outside the regime we studied theoretically. After correcting our bug, the initialization corresponds to a $T$ value of 2.6 - which numerically is closer to the regime of interest. And actually, if you look very closely, you can still see an initial phase with (smaller) oscillations - again coming from transient behavior due to large $T$. We apologize again for the issues with this figure!

---

### Official Review · Reviewer_T8Nt · 2022-10-26

**Confidence:** 3
**Correctness:** 4
**Technical Novelty And Significance:** 3
**Empirical Novelty And Significance:** Not applicable
**Recommendation:** 8

**Clarity, Quality, Novelty And Reproducibility:**

EoS is a recent and popular topic; there are several concurrent submissions also aiming to elucidate the EoS effect, in particular using somewhat related toy models (e.g. https://openreview.net/forum?id=p7EagBsMAEO, https://openreview.net/forum?id=nhKHA59gXz). However, I think that the present submission is sufficiently distinct from the others.

The appendix provides detailed proofs (I glanced over them but didn't check them) and some further comments on the models considered in the paper.

**Strength And Weaknesses:**

I like this paper. It is clearly written, gives a concise demonstration of the EoS effect on a toy model, proposes a natural general class of models shown to exhibit EoS, and the theoretical results are connected to the real world CIFAR data.

One very minor drawback that I see is that experiments with real world data are limited to a single network and data set.

**Summary Of The Paper:**

The paper studies the Edge-of-Stability effect using a second order regression model. The authors start with a scalar-valued quadratic model and show that in the case of two-dimensional input it has the EoS behavior in the non-symmetric eigenvalue scenario. After that the authors consider a more general, vector-valued second order regression model. They prove that in the high-dimensional limit and under appropriate random initialization, this model shows progressive sharpening at initialization. Then, the paper discusses the scaling at which the EoS behavior occurs. Finally, the paper shows experiments with CIFAR10 and relates them to preceding theoretical findings.

**Summary Of The Review:**

A good paper without any serious issues.

---

> ### Author Response · Authors · 2022-11-09
> **Response to reviewer T8Nt**
>
> We thank you for the review. We understand the criticism that only one real world dataset was used. This was in part due to the computational constraints in computing quantities like the curvature and the NTK. We have included a more detailed analysis of a 2-class CIFAR dataset in Appendix D.2 of the new draft, which may be of interest to the reviewer.

---

### Official Review · Reviewer_7fHN · 2022-10-27

**Confidence:** 3
**Clarity, Quality, Novelty And Reproducibility:** it is mostly clear.
**Correctness:** 3
**Technical Novelty And Significance:** 4
**Empirical Novelty And Significance:** 4
**Recommendation:** 6

**Strength And Weaknesses:**

Strength: I believe the authors tackle a quite important open problem related to SGD and the key message of looking at quadratic regression is valuable.
Weakness: one of those papers you need to sit down and parse the paper multiple times to get the intuition of the technical detail part.

**Summary Of The Paper:**

This paper made an effort towards understanding the edge of stability phenomenon. I think their major observation is that second order approximation is needed to analyze the gradient flow (the prediction is a quadratic function so the loss is a 4-th order function). Then they also propose a simplified quadratic regression model to capture the edge of stability behavior.

I feel the paper is answering a quite important question but I am unable to get the big picture intuition --- the only two messages i got was (i) looking at 2nd order approximation is the right way to explain edge of stability, and (ii) it requires very heavy computation to even get some simplified case out.

**Summary Of The Review:**

it is a strong result but I have low confidence.

---

> ### Author Response · Authors · 2022-11-09
> **Big picture intuition**
>
> Thank you for the review. The main intuition from the paper is that the EOS behavior shows up even in the simple quadratic regression model; in simplified models we can even quantitatively understand the dynamics as coming from feedback between the curvature and the displacement from the minimum. In the more general case, the dynamics is more complicated but seems to be related to interaction of the large eigenmode components of z with the Q tensor.

---

### Official Review · Reviewer_GxcF · 2022-10-27

**Confidence:** 1
**Clarity, Quality, Novelty And Reproducibility:** The paper study some fundamental prop…
**Correctness:** 4
**Technical Novelty And Significance:** 2
**Empirical Novelty And Significance:** 2
**Recommendation:** 6

**Strength And Weaknesses:**

Strength
[+] The paper studied a second order regression models and analyze the progressive sharpening and ESO in details.

Weakness
[-] The paper takes time to follow and some of the organization could be better.
(a) \title{z} is used before the definition, before and after equation (3)
(b) curvature (JJ^\top)  could be defined after the definition of JJ^\top after (3)


**Summary Of The Paper:**

The paper consider a class of predictive models that are quadratic in parameters (equation 1 and 19),  and prove progressive sharpening (section 2.1) and edge of stability (ESO, section 2.2) in two dimension.  The paper also claims that the two properties could be general property in high dimensional non-linear models.

**Summary Of The Review:**

The paper study some fundamental property of second order regression.

---

> ### Author Response · Authors · 2022-11-09
> **Thanks for the review**
>
> We've incorporated your comments in the new draft.

---

### Official Review · Reviewer_saA5 · 2022-11-01

**Confidence:** 4
**Correctness:** 3
**Technical Novelty And Significance:** 3
**Empirical Novelty And Significance:** 2
**Recommendation:** 5

**Clarity, Quality, Novelty And Reproducibility:**

The work is original and of high quality. However, the paper involves a lot of mathematical jargon, which makes it difficult for the reader to understand the main message behind different theorems and experiments. I have pointed out some in the previous section.


**Strength And Weaknesses:**

The major strength of the paper is to show that second-order NTKs can show the Edge of stability behavior close to initialization, under proper conditions.


However, I feel the main messages of sections 2.2.2, 3.2, and 3.3 are not clear at all. I have the following questions.
a) How should one be reading the plot in Figure 2 (middle)?

When the authors mention "For small, the dynamics show the distinct phases described in (Li et al., 2022): an initial increase in $T(0)$, a slow increase in $\tilde{z}$, then a decrease in $T(0)$, and finally a slow decrease of $\tilde{z}$ while $T(0)$ remains near 2 (Figure 2, middle)", how should one read this phenomenon in Figure 2 (middle)?

Moreover, what are the blue lines in figure 2 (middle)?

b) How does the result of theorem 3.1 imply progressive sharpening? The theorem statement simply says that the expected time derivative of $\lambda_{\max}$ is $0$ at initialization, while the expected double derivative of $\lambda_{\max}$ depends on the initial scale of output. How does this imply that $\lambda_{\max}$ will increase to $2/\eta$ after few steps?

Furthermore, the jacobian $J = G + Q(\theta, .)$, how do the authors initialize $J$ with variance $\sigma_J^2$? Or are the authors initializing both $G$ and $Q$ such that the condition on $J$ holds true? Same question for $z$, how do the authors make sure $z$ has variance $\sigma_z$?


c) In theorem 2.1, the authors require the second eigenvalue to be much smaller than $1$, otherwise, the bound on $\lim_{t \to \infty} \lambda_{\max}$ gets very loose. Can the authors comment on whether such a big gap between the eigenvalues is necessary for EoS to hold in practice? Or do the authors need such a gap to make sure the EoS occurs near initialization?

Furthermore, Cohen et al. [2020] observe that the EoS phenomenon is robust to the scale of the learning rate. However, in theorem 2.1, we require the learning rate to stay in a range $[\eta_1, \eta_2]$. Can the authors comment on this necessity in the proof?


d) Is the main message behind section 3.3 that we need the second-order term in NTK for the EoS phenomena? Can the authors comment on how the ratio D/P matters affect theorem 3.1? How will the plots in Figure 3 and Figure 4 change with different D/P values?

Furthermore, in theorem 2.1, we needed a large gap between the top two eigenvalues of Q. Is that somehow related to $\sigma_z$ and $\sigma_J$?


**Summary Of The Paper:**

The paper aims to understand the recently discovered "Edge of Stability" phenomena in a second-order regression model. The authors consider second-order NTK models and show for simple settings, under proper conditions,  one can observe progressive sharpening close to initialization. Finally, the authors simulate the simple models to observe the EoS phenomena.

**Summary Of The Review:**

Overall, my scores are on the borderline. The mathematical jargons in the paper make it really difficult to understand the main message (and the underlying idea) behind the important sections. Please find my questions above.

---

> ### Author Response · Authors · 2022-11-09
> **Response to Reviewer saA5**
>
> We thank the reviewer for their detailed comments. We will quote the relevant sections of the review below and respond to each point individually.
>
> _I feel the main messages of sections 2.2.2, 3.2, and 3.3 are not clear at all._
>
> We’re sorry for the confusion in messaging! The main messages are as follows:
>
> 2.2.2 - When eigenvalues of Q are imbalanced, a 2-parameter model on 1-datapoint can provably show EOS behavior - and we can quantify the distance to the edge.
>
> 3.2 - The quadratic regression model, when randomly initialized, tends to increase curvature at early times.
>
> 3.3 - When a quadratic regression model is initialized with sufficiently large z and lr, the dynamics is non-linear and displays EOS behavior robustly.
>
> _a) How should one be reading the plot in Figure 2 (middle)?_
>
> Figure 2, middle: the blue lines are the trajectories of the simple model (1 datapoint, 2 parameters) in the z-T space, for various initializations. They are plotted every-other iterate so the focus is on the overall trend, and not the oscillations. The trajectories start at small z, T>2, (now marked with an x) and end at z = 0, T<2 (but close). The increase and decrease phases can be read off from the trajectories, but the timescales cannot.
>
> _b) How does the result of theorem 3.1 imply progressive sharpening? The theorem statement simply says that the expected time derivative of  is  at initialization, while the expected double derivative of  depends on the initial scale of output. How does this imply that  will increase to $2/\eta$ after few steps?_
>
> The theorem implies that the maximum eigenvalue will tend to increase at early times. It doesn’t guarantee that the eigenvalue will reach the edge of stability after a few steps. The main point is to establish that our simplified model, like many real neural networks has a regime where the maximum eigenvalue is increasing (progressive sharpening). As shown empirically in Cohen et. al., progressive sharpening is a necessary (but not sufficient) condition for EOS behavior to emerge. Our theorem helps solidify the empirical finding that the learning curves display progressive sharpening (Figure 12, Appendix C in latest version).
>
> _Furthermore, the jacobian $J$, how do the authors initialize $J$ with variance $\sigma_J^2$? Or are the authors initializing both $G$ and $Q$ such that the condition on $J$ holds true? Same question for $z$_
>
> We initialize z and J directly, and then use Equations 25 and 26 to carry out the dynamics. We chose this direct approach rather than the indirect approach of initializing \theta, G, and Q such that z and J have the desired statistics.
>
> _c) In theorem 2.1, the authors require the second eigenvalue to be much smaller than $1$, otherwise, the bound on  gets very loose. Can the authors comment on whether such a big gap between the eigenvalues is necessary for EoS to hold in practice? Or do the authors need such a gap to make sure the EoS occurs near initialization?”_
>
> The bound is loose because for larger epsilon, the final curvature is no longer close to the EOS. The gap is needed to get the behavior in our one datapoint, 2 parameter model. We believe that the ratio of eigenvalues is needed to achieve what is known in dynamical systems studies as a “separation of timescales” - to allow a fast process (the equilibration of T or more precisely y to the nullcline) to coexist with a slow process (the slow decay of z). In the high dimensional case, our experiments indicate that this is not necessary; the separation of timescales may come from another source (for example, the projection of Q in the top eigendirection versus a random one).

---

> > ### Author Response · Authors · 2022-11-09
> > **Response (continued)**
> >
> > _Furthermore, Cohen et al. [2020] observe that the EoS phenomenon is robust to the scale of the learning rate. However, in theorem 2.1, we require the learning rate to stay in a range . Can the authors comment on this necessity in the proof?_
> >
> > Even in the experiments of Cohen et. al., there is a window of learning rates that display EOS phenomenology. Very small learning rates lead to behavior similar to GF; very large learning rates cause instability which can’t be recovered from. Similarly, in Theorem 2.1 there is a range of learning rates over which the phenomenology exists. We note that the range in the theorem is conservative, and also note that the question of computing the range for different models is an open one.
> >
> > _d) Is the main message behind section 3.3 that we need the second-order term in NTK for the EoS phenomena? Can the authors comment on how the ratio D/P matters affect theorem 3.1? How will the plots in Figure 3 and Figure 4 change with different D/P values?”_
> >
> > This is correct - the second order term is in fact _sufficient_ to obtain EOS behavior. We believe that as D/P becomes small (overparameterized regime), while D and P are both large, that progressive sharpening becomes more robust. We have added some plots in Appendix C.3 in the latest version which provides empirical evidence for this hypothesis. Understanding this phenomenology quantitatively is an avenue for future work.
> >
> > _Furthermore, in theorem 2.1, we needed a large gap between the top two eigenvalues of Q. Is that somehow related to $\sigma_J$ and $\sigma_z$?_
> >
> > The gap in Theorem 2.1 is needed in order to make the phenomenology robust in low dimensions; however the initialization of $z$ is related to $\sigma_z$ and $T$ related to $\sigma_J$. For example, if $T$ and $z$ are both initialized to be small, then there is not sufficient non-linearity to achieve EOS.
> >
> > Let us know if you have any followup questions; the new draft will be posted within an hour of this comment.

---

> > > ### Author Response · Authors · 2022-11-15
> > > **Anything else to clear up?**
> > >
> > > Let us know if there's anything else we can resolve. Additionally, if there are any further suggestions to improve the clarity of the draft, we would be happy to implement them. Thanks!

---

### Author Response · Authors · 2022-11-09
**Updated draft uploaded**

We thank all the reviewers for their thoughtful comments. We have uploaded an update draft. In addition to minor revisions based on reviewer comments, we have added:

Edits to figures 1 and 2 to show the initial values for trajectories.

Appendix A.1: relationship between Hessian spectrum and NTK spectrum.

Appendix B.6/Figure 11: explains the relationship between Figure 1 and Figure 2.

Appendix C.3/Figure 14: dependence of EOS on $D$ and $P$.

Let us know if there are any other points we can clear up!

---

### Decision · Program_Chairs · 2023-01-20

**Decision:**

Reject

**Justification For Why Not Higher Score:**



**Justification For Why Not Lower Score:**



**Metareview: Summary, Strengths And Weaknesses:**

The submission is about developing an understanding of the recently observed progressive sharpening and edge of stability phenomena. The authors focus on second-order regression models (i.e. models which are quadratic in their parameters) with two-dimensional input to study the question analytically, and show empirically that these two properties could generally be present in high-dimensional non-linear models.

While the overall question raised can potentially have some impact, unfortunately, the submission in its current form would require significant revision before publication: 'the main messages are not clear at all', the reviewers were 'unable to get the big picture' and to decode the key messages of the work; my own reading also confirms this rather bumpy experience.